# WHERE HAVE YOU BEEN? A STUDY OF PRIVACY RISK FOR POINT-OF-INTEREST RECOMMENDATION

## ABSTRACT

As location-based services (LBS) have grown in popularity, the collection of human mobility data has become increasingly extensive to build machine learning (ML) models offering enhanced convenience to LBS users. However, the convenience comes with the risk of privacy leakage since this type of data might contain sensitive information related to user identities, such as home/work locations. Prior work focuses on protecting mobility data privacy during transmission or prior to release, lacking the privacy risk evaluation of mobility data-based ML models. To better understand and quantify the privacy leakage in mobility data-based ML models, we design a privacy attack suite containing data extraction and membership inference attacks tailored for point-of-interest (POI) recommendation models, one of the most widely used mobility data-based ML models. These attacks in our attack suite assume different adversary knowledge and aim to extract different types of sensitive information from mobility data, providing a holistic privacy risk assessment for POI recommendation models. Our experimental evaluation using two real-world mobility datasets demonstrates that current POI recommendation models are vulnerable to our attacks. We also present unique findings to understand what types of mobility data are more susceptible to privacy attacks. Finally, we evaluate defenses against these attacks and highlight future directions and challenges.

## 1 INTRODUCTION

With the development and wide usage of mobile and wearable devices, large volumes of human mobility data are collected to support location-based services (LBS) such as traffic management (Bai et al., 2020; Lan et al., 2022), store location selection (Liu et al., 2017), and point-of-interest (POI) recommendation (Sun et al., 2020; Yang et al., 2022). In particular, POI recommendation involves relevant POI suggestions to users for future visits based on personal preferences using ML techniques (Islam et al., 2020) and is widely deployed. POI recommendation has been integrated into popular mapping services such as Google Maps to assist users in making informed decisions about the next destination to visit. However, mobility data collection raises privacy concerns as it can leak users' sensitive information such as their identities (Blumberg & Eckersley, 2009).

Although there are a significant number of studies (Gedik & Liu, 2005; Krumm, 2007a; Andrés et al., 2013; Shokri et al., 2013) on mobility data privacy, the existing research primarily focuses on analyzing attacks and evaluations within the context of LBS data transmission and release processes. For example, previous studies have demonstrated the linkages of mobility data from various side channels, including social networks (Henne et al., 2013; Hassan et al., 2018), open-source datasets (Gambs et al., 2014; Powar & Beresford, 2023), and network packets (Jiang et al., 2007; Vratonjic et al., 2014). The linkages between these side channels can lead to the identification of individuals. As a result, efforts to protect mobility data have primarily concentrated on data aggregations and releases (Gedik & Liu, 2005; Meyerowitz & Roy Choudhury, 2009; Bordenabe et al., 2014). These studies neglect the risk of adversaries extracting sensitive attributes or properties from the ML models that use mobility data for training, which are inherently susceptible to inference attacks (Shokri et al., 2017; Carlini et al., 2022).

Previous inference attacks have mainly focused on ML models trained with image and text data (Shokri et al., 2017; Fredrikson et al., 2015a; Carlini et al., 2019). While texts and images capture

static information, mobility data contain multimodal spatial and temporal information which provides insights into individuals' movements and behaviors over time. However, this unique characteristic of mobility data makes it vulnerable to potential adversaries who can obtain mobility data to infer the identity or trace the user's behavior (De Montjoye et al., 2015; Liu et al., 2018). Furthermore, existing defense mechanisms (Abadi et al., 2016; Shi et al., 2022a;b) have mainly been tested on models trained with image or text data, making their effectiveness uncertain when applied to POI recommendation models.

In this paper, we design a privacy attack suite to study the privacy leakage in POI recommendation models trained with mobility data. We are selecting POI recommendation models for our study due to their widespread usage in LBS (Wang et al., 2023). Specifically, the privacy attack suite contains data extraction and membership inference attacks to assess the privacy vulnerabilities of POI recommendation models at both location and trajectory levels. In contrast to privacy attacks for image and text data, the attacks in our attack suite are uniquely adapted for mobility data and aim to extract different types of sensitive information by assuming different adversary knowledge.

We perform experiments on three representative POI recommendation models trained on two mobility benchmark datasets. We demonstrate that POI recommendation models are vulnerable to our designed data extraction and membership inference attacks. We further provide an in-depth analysis to understand what factors affect the attack performance and contribute to the effectiveness of the attacks. Based on our analysis, we discover that the effect of data outliers exists in privacy attacks against POI recommendations, making training examples with certain types of users, locations, and trajectories particularly vulnerable to the attacks in the attack suite. We further test several existing defenses and find that they do not effectively thwart our attacks with negligible utility loss, which calls for better methods to defend against our attacks.

**Contributions** (1) We propose a unique privacy attack suite for POI recommendation models. To the best of our knowledge, our work is the first to comprehensively evaluate the privacy risks in POI recommendation models using inference attacks. (2) We conduct extensive experiments demonstrating that POI recommendation models are vulnerable to data extraction and membership inference attacks in our attack suite. (3) We provide an in-depth analysis to understand what unique factors in mobility data make them vulnerable to privacy attacks, test defenses against the attacks, and identify future directions and challenges in this area.

## 2 BACKGROUND

### 2.1 POINT-OF-INTEREST RECOMMENDATION

POI recommendation has recently gained much attention[1] due to its importance in many business applications (Islam et al., 2020), such as user experience personalization and resource optimization. Initially, researchers focused on feature engineering and algorithms such as Markov chain (Zhang et al., 2014; Chen et al., 2014), matrix factorization algorithms (Lian et al., 2014; Cheng et al., 2013), and Bayesian personalized ranking (He et al., 2017; Zhao et al., 2016) for POI recommendation. However, more recent studies have shifted their attention towards employing neural networks like RNN (Liu et al., 2016; Yang et al., 2020), LSTM (Kong & Wu, 2018; Sun et al., 2020), and self-attention models (Luo et al., 2021; Lian et al., 2020). Neural networks can better learn from spatial-temporal correlation in mobility data (e.g., check-ins) to predict users' future locations and thus outperform other POI recommendation algorithms by a large margin. Meanwhile, this could introduce potential privacy leakage. Thus, we aim to design an attack suite to measure the privacy risks of neural-network-based POI recommendations systematically.

We first provide the basics of POI recommendations and notations used throughout this paper. Let $\mathcal{U}$ be the user space, $\mathcal{L}$ be the location space, and $\mathcal{T}$ be the timestamp space. A POI recommendation model takes the observed trajectory of a user as input and predicts the next POI that will be visited, which is formulated as $f_\theta : \mathcal{U} \times \mathcal{L}^n \times \mathcal{T}^n \to \mathbb{R}^{|\mathcal{L}|}$. Here, the length of the input trajectory is $n$. We denote a user by its user ID $u \in \mathcal{U}$ for simplicity. For an input trajectory with $n$ check-ins, we denote its trajectory sequence as $x_T^{0:n-1} = \{(l_0, t_0), \ldots, (l_{n-1}, t_{n-1})\}$, where $l_i \in \mathcal{L}$ and $t_i \in \mathcal{T}$ indicate the POI

---

[1]From 2017 to 2022, there are more than 111 papers on POI recommendation built upon mobility data collected by location service providers (Wang et al., 2023).

location and corresponding time interval of $i$-th check-in. Also, the location sequence of this trajectory is denoted as $x_L^{0:n-1} = \{l_0, \ldots, l_{n-1}\}$. The POI recommendation model predicts the next location $l_n$ (also denoted as $y$ by convention) by outputting the logits of all the POIs. Then, the user can select the POI with the highest logit as its prediction $\hat{y}$, where $\hat{y} = \arg\max f_\theta(u, x_T^{0:n-1})$. Given the training set $D_{\text{tr}}$ sampled from an underlying distribution $\mathcal{D}$, the model weights are optimized to minimize the prediction loss on the overall training data, i.e., $\min_\theta \frac{1}{|D_{\text{tr}}|} \sum_{(u, x_T^{0:n-1}, y) \in D_{\text{tr}}} \ell(f_\theta(u, x_T^{0:n-1}), y)$, where $\ell$ is the cross-entropy loss, i.e., $\ell(f_\theta(u, x_T^{0:n-1}), y) = -\log(f_\theta(u, x_T^{0:n-1}))_y$. The goal of the training process is to maximize the performance of the model on the unseen test dataset $D_{te} \in \mathcal{D}$, which is drawn from the same distribution as the training data. During inference, this prediction $\hat{y}$ is then compared to the next real location label $l_n$ to compute the prediction accuracy. The performance evaluation of POI recommendation models typically employs metrics such as top-$k$ accuracy (e.g., $k = 1, 5, 10$).

## 2.2 THREAT MODELS

Next, we introduce the threat models of our attacks (see also Table 2 in the appendix for clarity).

**Adversary Objectives** To understand the potential privacy leakage of training data in POI recommendation models, we design the following four attacks based on the characteristics of the mobility data for POI recommendation, namely *common location extraction* (LOCEXTRACT), *training trajectory extraction* (TRAJEXTRACT), *location-level membership inference attack* (LOCMIA), and *trajectory-level membership inference attack* (TRAJMIA).

These four attacks aim to extract or infer different sensitive information about a user in the POI recommendation model training data. LOCEXTRACT focuses on extracting a user's most frequently visited location; TRAJEXTRACT aims to extract a user's location sequence with a certain length given a starting location; LOCMIA is to infer whether a user has been to a location and used for training; TRAJMIA is to infer the training membership of a trajectory sequence.

**Adversary Knowledge** For all attacks, we assume the attacker has access to the query interface of the victim model. Specifically, the attacker can query the model with the target user to attack any location and obtain the model output logits. This assumption is realistic in two scenarios: (1) A malicious third-party entity is granted access to the POI model query API hosted by the model owner (e.g., location service providers) for specific businesses such as personalized advertisement. This scenario is well-recognized by Shubham Sharma (2022); Mike Boland (2021); Xin et al. (2021). (2) The retention period of the training data expires. Still, the model owner keeps the model and an adversary (e.g., malicious insider of location service providers) can extract or infer the sensitive information using our attack suite, even if the training data have been deleted. In this scenario, the model owner may violate privacy regulations such as GDPR (EU, 2018).

Depending on different attack objectives, the adversary also possesses different auxiliary knowledge. In particular, for TRAJEXTRACT, we assume the attacker can query the victim model with a starting location $l_0$ that the target user visited. This assumption is reasonable because an attacker can use real-world observation (Vicente et al., 2011b; Srivatsa & Hicks, 2012b), LOCEXTRACT, and LOCMIA as cornerstones. As for LOCMIA and TRAJMIA, we assume the attacker has access to a shadow dataset following the standard settings of membership inference attacks (Shokri et al., 2017; Carlini et al., 2022).

## 3 ATTACK SUITE

## 3.1 DATA EXTRACTION ATTACKS

Our data extraction attacks are rooted in the idea that victim models display varying levels of memorization in different subsets of training data. By manipulating queries to the victim models, the attacker can extract users' locations or trajectories that these victim models predominantly memorize.

**LOCEXTRACT** The common location extraction attack (LOCEXTRACT) aims to extract a user's most frequently visited location in the victim model training, i.e.,

$$\text{LOCEXTRACT}(f_\theta, u) \rightarrow \hat{l}_{top1}, \ldots, \hat{l}_{topk}.$$

The attack takes the victim model $f_\theta$ and the target user $u$ as the inputs and generates $k$ predictions $\hat{l}_{top1}, \ldots, \hat{l}_{topk}$ to extract the most frequently visited location of user $u$. The attack is motivated by our key observation that when querying POI recommendation models with a random location, POI recommendation models "over-learn" the user's most frequently visited locations. For example, we randomly choose 10 users and query to the victim model using 100 randomly selected locations. Of these queries, 32.5% yield the most frequent location for the target user. Yet, these most common locations are present in only 18.7% of these users' datasets.

In LOCEXTRACT, we first generate a set of different random inputs for a specific user and use them to make iterative queries to the victim model. Each query returns the prediction logits with a length of $|\mathcal{L}|$ outputted by the victim model. The larger the logit value, the more confident the model is in predicting the corresponding location as the next POI. Therefore, by iterating queries to the model given a target user and aggregating the logit values of all queries, the most visited location is more likely to have a large logit value after aggregation. Here, we use a soft voting mechanism, i.e., averaging the logit values of all queries, for the aggregation function (see also Sec. D for the comparison with different aggregation functions). With the resulting mean logits, we output the top-$k$ locations with $k$ largest logit values as the attack results. Algorithm 1 gives the outline of LOCEXTRACT. Though the attack is straightforward, it is effective and can be a stepping stone for TRAJEXTRACT in our attack suite.

**TRAJEXTRACT**    Our training trajectory extraction attack (TRAJEXTRACT) aims to extract the location sequence $x_L^{0:n-1} = \{l_0, \ldots, l_{n-1}\}$ in a training trajectory of user $u$ with a length of $n$ from the victim model $f_\theta$. Formally,

$$\text{TRAJEXTRACT}(f_\theta, u, l_0, n) \to \hat{x}_{L_0}^{0:n-1}, \ldots, \hat{x}_{L_\beta}^{0:n-1},$$

where $\hat{x}_{L_0}^{0:n-1}, \ldots, \hat{x}_{L_\beta}^{0:n-1}$ indicate the top-$\beta$ extracted location sequences by the attack.

The key idea of the training trajectory extraction attack is to identify the location sequence with the lowest log perplexity, as models tend to demonstrate lower log perplexity when they see trained data. We denote log perplexity as:

$$\text{PPL}_{f_\theta}(u, x_T^{0:n-1}) = -\log \text{Pr}_{f_\theta}(u, x_T^{0:n-1}) = -\sum_{i=0}^{n-1} \log \text{Pr}_{f_\theta}(u, x_T^{0:i-1}),$$

where $\text{Pr}_{f_\theta}(\cdot)$ is the likelihood of observing $x_T^{0:n-1}$ with user $u$ under the victim model $f_\theta$. In order to get the lowest log perplexity of location sequences with a length of $n$, we have to enumerate all possible location sequences. However, in the context of POI recommendation, there are $\mathcal{O}(|\mathcal{L}|^{n-1})$ possible location sequences for a given user. $|\mathcal{L}|$ equals the number of unique POIs within the mobility dataset and can include thousands of options. Thus, the cost of calculating the log perplexity of all location sequences can be very high. To this end, we use beam search to extract the location sequences with both time and space complexity $\mathcal{O}(|\mathcal{L}| \times n \times \beta)$, where $\beta$ is the beam size. In particular, to extract a trajectory of length $n$, we iteratively query the victim model using a set of candidate trajectories with a size of $\beta$ and update the candidate trajectories until the extraction finishes. As highlighted in the prior work (Fan et al., 2018), when using beam search to determine the final outcome of a sequential neural network, there is a risk of generating non-diverse outputs and resembling the training data sequence. However, in our scenario, this property can be leveraged as an advantage in TRAJEXTRACT, as our primary objective revolves around extracting the training location sequence with higher confidence. As a final remark, both LOCEXTRACT and TRAJEXTRACT need a query timestamp to query the victim model, and we will show the effects of the timestamp in our experiments. Algorithm 2 in Appendix A gives the detailed steps of TRAJEXTRACT

## 3.2   MEMBERSHIP INFERENCE ATTACKS

Membership inference attack (MIA) originally aims to determine whether a target data sample is used in the model training. In our case, we extend the notion to infer whether certain sensitive information (e.g., user-location pair $(u, l)$ and trajectory sequence $(u, x_T)$) of the user's data is involved in the training of the victim model $f_\theta$, which can be formulated as follows:

$$\text{MIA}(f_\theta, X_{target}, D_s) \to \{\text{member}, \text{nonmember}\},$$

where $X_{target}$ represents the target sensitive information ($X_{target} = (u, l)$ in LocEXTRACT and $X_{target} = (u, x_T)$ in TRAJEXTRACT), and $D_s$ is the shadow dataset owned by the adversary.

To effectively infer the membership of a given $X_{target}$, we adapt the state-of-the-art membership inference attack – likelihood ratio attack (LiRA) (Carlini et al., 2022) to the context of POI recommendation. The key insight of LiRA is that the model parameters trained with $X_{target}$ differ from those trained without it, and by conducting a hypothesis test on the distributions of model parameters, we can identify if the victim model is trained with the $X_{target}$ or not. LiRA consists of three steps: (1) shadow model training, (2) querying the victim model and shadow models using $X_{target}$, and (3) conducting a hypothesis test to infer the membership of the $X_{target}$ using the query results. Due to the space limit, we defer the details of LiRA to Appendix A.

**LocMIA** In this attack, the adversary aims to determine whether a given user $u$ has visited a location $l$ in the training data. However, it is not feasible to directly apply LiRA to LocMIA as the victim model takes the trajectory sequences as inputs, but the adversary only has a target location without the needed sequential context. In particular, LocMIA needs the auxiliary inputs to calculate the membership confidence score since this process cannot be completed only using $X_{target} = (u, l)$. This attack is a stark contrast to MIA for image/text classification tasks where the $X_{target}$ itself is sufficient to compute the membership confidence score.

To this end, we design a spatial-temporal model query algorithm (Algorithm 3 in Appendix A) to tailor LiRA to LocMIA and optimize membership confidence score calculation. The idea behind the algorithm is that if a particular user has been to a certain POI location, the model might "unintentionally" memorize its neighboring POI locations and the corresponding timestamp in the training data. Motivated by this, each time we query the models (e.g., the victim and shadow models), we generate $n_l$ random locations and $n_t$ fixed-interval timestamps. To obtain stable and precise membership confidence scores, we first average the corresponding confidence scores at the target location by querying with $n_l$ locations at the same timestamp. While the adversary does not possess the ground truth timestamp linked with the target POI for queries, the adversary aims to mimic a query close to the real training data. To achieve this, we repeat the same procedure of querying different locations for $n_t$ timestamps and take the maximum confidence scores among the $n_t$ averaged confidence scores as the final membership inference score for the target example. Algorithm 4 gives the outline of LiRA in terms of LocMIA, and the lines marked with red are specific to LocMIA.

**TRAJMIA** The attack aims to determine whether a trajectory is used in the training data of the victim model. Unlike LocMIA, $X_{target} = (u, x_T)$ suffices to calculate the membership confidence score in LiRA, and we do not need any auxiliary inputs. To fully leverage information of the target example querying the victim model and improve the attack performance, we also utilize the $n - 2$ intermediate outputs and the final output from the sequence $x_T$ with a length of $n$ to compute the membership confidence score, i.e., we take the average of all $n - 1$ outputs. This change improves the attack performance as the intermediate outputs provide additional membership information for each point in the target trajectory. The purple lines in Algorithm 4 highlight steps specific to TRAJMIA.

### 3.3 PRACTICAL IMPLICATIONS OF THE ATTACK SUITE

Our attack suite is designed as an integrated framework focusing on the basic units of mobility data – locations and trajectories. It contains two prevalent types of privacy attacks: data extraction and membership inference attacks. Each attack in our attack suite targets a specific unit of mobility data and could serve as a privacy auditing tool (Jagielski et al., 2020). They can also be used to infer additional sensitive information in mobility data: LocEXTRACT extracts a user's most common location locations and combine the POI information to infer the users' home and work locations; TRAJEXTRACT can be further used to infer user trajectories and identify trip purposes by analyzing the POIs visited during a journey (Meng et al., 2017); LocMIA can determine the membership of multiple POIs, thereby facilitating the inference of a user's activity range and social connections in Cho et al. (2011); Ren et al. (2023); Finally, TRAJMIA infers if a user's trajectory is in a training dataset can serve as an auditing tool to examine the privacy leakage of a given model by assuming a worst-case adversary.

## 4 EXPERIMENTS

We empirically evaluate the proposed attack suite to answer the following research questions: (1) How are the proposed attacks performed in extracting or inferring the sensitive information about

the training data for POI recommendation (Sec. 4.2.1)? (2) What unique factors (e.g., user, location, trajectory) in mobility data correlate with the attack performance (Sec. 4.2.2)? (3) Are the existing defenses effective against the proposed attacks (Sec. 4.2.3)? In Appendix D, we also study the effects of model training and attack (hyper)parameters on attack performance.

## 4.1 EXPERIMENTAL SETUP

We briefly describe the datasets, models, and evaluation metrics used in our experiments. Due to the space limit, we defer the details of dataset statistics and descriptions, data preprocessing steps, and (hyper)parameters of training and attacks to Appendix A.

**Datasets** Following the literature (Yang et al., 2022; Kong & Wu, 2018), we comprehensively evaluate four privacy attacks on two POI recommendation benchmark datasets: FourSquare (4SQ) (Yang et al., 2014) and GOWALLA (Cho et al., 2011).

**Models** We experiment with three representative POI recommendation models, including GET-NEXT (Yang et al., 2022), LSTPM (Sun et al., 2020), and RNN (Wang et al., 2021). Note that GETNEXT and LSTPM are the state-of-the-art POI recommendation methods based on the transformer and hierarchical LSTM, respectively. We also include RNN since it is a commonly used baseline for POI recommendation.

**Evaluation Metrics** We use the top-$k$ extraction attack success rate (ASR) to evaluate the effectiveness of data extraction attacks. For LOCEXTRACT, the top-$k$ ASR is defined as $|U_{\text{extracted}}|/|\mathcal{U}|$, where $U_{\text{extracted}}$ is the set of users whose most visited locations are in the top-$k$ predictions outputted by our attack; For TRAJEXTRACT the top-$k$ ASR is |correct extractions|/|all $(u, l_0)$ pairs|, where correct extractions are $(u, l_0)$ pairs with top-$k$ extracted results matching an exact location sequence in the training data.

For LOCMIA and TRAJMIA, we utilize the commonly employed metrics for evaluating membership inference attacks, namely the area under the curve (AUC), average-case "accuracy" (ACC), and true positive rate (TPR) versus false positive rate (FPR) in the low-false positive rate regime. Our primary focus is the TPR versus FPR metric in the low-false positive rate regime because evaluating membership inference attacks should prioritize the worst-case privacy setting rather than average-case metrics, as emphasized in Carlini et al. (2022).

## 4.2 EXPERIMENTAL RESULTS AND ANALYSIS

### 4.2.1 ATTACK PERFORMANCE

Figures 1 and 2 visualize the attack performance of data extraction and membership inference attacks, respectively. In Figure 1, we observe that both LOCEXTRACT and TRAJEXTRACT can effectively extract users' most common locations and trajectories across various model architectures and datasets since the attack performance is significantly better than the random guess baseline, i.e., $1/|\mathcal{L}|$ (0.04% for LOCEXTRACT) and $1/|\mathcal{L}|^{n-1}$ ($10^{-8}$% for TRAJEXTRACT). Likewise, as shown in Figure 2, LOCMIA and TRAJMIA successfully

Table 1: The performance of victim models.

| Dataset | Model | Top-1 ACC | Top-10 ACC |
|---------|-------|-----------|------------|
| 4SQ | GETNEXT | 0.34 | 0.71 |
| | LSTPM | 0.25 | 0.67 |
| | RNN | 0.24 | 0.68 |
| GOWALLA | GETNEXT | 0.16 | 0.48 |
| | LSTPM | 0.15 | 0.39 |
| | RNN | 0.10 | 0.26 |

determine the membership of a specific user-location pair or trajectory, significantly outperforming the random guess baseline (represented by the diagonal line in both figures).

The attack performance also demonstrates that trajectory-level attacks are significantly more challenging than location-level attacks, evident from the better performance of LOCEXTRACT and LOCMIA compared to TRAJEXTRACT and TRAJMIA for data extraction and membership inference. We suspect this is because POI recommendation models are primarily designed to predict a single location. In contrast, our trajectory-level attacks aim to extract or infer a trajectory encompassing multiple consecutive locations. The experiment results also align with the findings that longer trajectories are less vulnerable to our attacks (see Figures 14 and 17 in Appendix D).

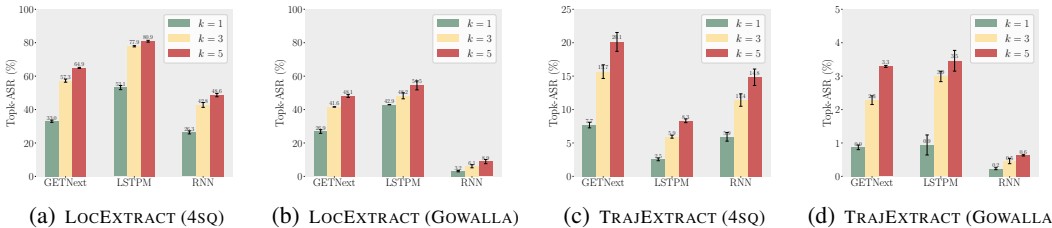

Figure 1: Attack performance of data extraction attacks (LOCEXTRACT and TRAJEXTRACT) on three victim models and two mobility datasets.

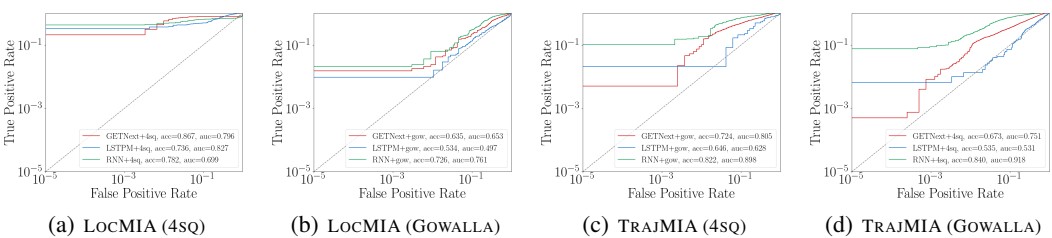

Figure 2: Attack performance of (LOCMIA and TRAJMIA) on three victim models and two POI recommendation datasets. The diagonal line indicates the random guess baseline.

The attack performance also differs across different model architectures and datasets. Combined with the model performance of the victim model in Table 1, we see a general trend of privacy-utility trade-off in POI recommendation models: with better victim model performance comes better attack performance in general. However, this trend does not hold in some cases. For example, the MIA performance against RNN is sometimes better than GETNEXT and LSTPM performances. This might be because GETNEXT and LSTPM improve upon RNN by better leveraging spatial-temporal information in the mobility datasets. However, the adversary cannot use the exact spatial-temporal information in shadow model training since the adversary cannot access that information. Even though our attacks adapt LiRA to utilize spatial-temporal information (see Appendix D.3 for more results), there is room for improvement in future research.

### 4.2.2 FACTORS IN MOBILITY DATA THAT MAKE IT VULNERABLE TO THE ATTACKS

Prior research demonstrates that data outliers are the most vulnerable examples to privacy attacks (Carlini et al., 2022; Tramèr et al., 2022) in image and text datasets. However, it is unclear whether the same conclusion holds in mobility data and what makes mobility data as data outliers. To this end, we investigate which factors of the mobility datasets influence the attack's efficacy. In particular, we collect aggregate statistics of mobility data from three perspectives: user, location, and trajectory. We analyze which factors in these three categories make mobility data vulnerable to our attacks. We defer the details of selecting the aggregate statistics and the list of selected aggregate statistics in our study in Appendix C.1. Our findings are as follows:

- For LOCEXTRACT, we do not identify any meaningful pattern correlated with its attack performance. We speculate that a user's most common location is not directly related to the aggregate statistics we study.

- For TRAJEXTRACT, our findings indicate that *users who have visited fewer unique POIs* are more vulnerable to this attack, as referenced in Figure 6 in Appendix C. This can be explained by the fact that when users have fewer POIs, the model is less uncertain in predicting the next location due to the reduced number of possible choices that the model memorizes.

- For LOCMIA, as shown in Figures 4(a) and 4(a), we find that *locations visited by fewer users or have fewer surrounding check-ins* are more susceptible to LOCMIA. We believe this is because those locations shared with fewer users or surrounding check-ins make them training data outliers.

- For TRAJMIA, *users with fewer total check-ins* (Figure 3(a)), *unique POIs* (Figure 3(b)), and *fewer or shorter trajectories* (Figures 3(c) and 3(d)) are more susceptible. In Figures 5(a) and 5(b), we also see that *trajectories intercepting less with others or with more check-ins* are more vulnerable to TRAJMIA. We believe these user-level and trajectory-level aggregate statistics make the target examples data outliers.

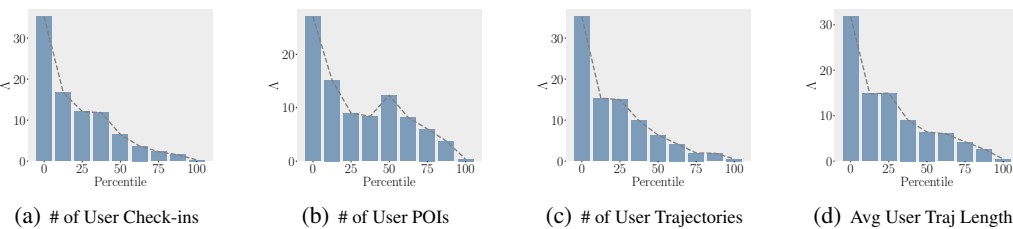

(a)  # of User Check-ins     (b)  # of User POIs     (c)  # of User Trajectories     (d)  Avg User Traj Length

Figure 3: How user-level aggregate statistics are related to TRAJMIA. *x-axis:* Percentile categorizes users/locations/trajectories into different groups according to their feature values. *y-axis:* $\Lambda$ indicates the (averaged) likelihood ratio of training trajectories/locations being the member over non-member from the hypothesis test for each group, with a higher value indicating the larger vulnerability. The users with fewer total check-ins, fewer unique POIs, and fewer or shorter trajectories are more vulnerable to TRAJMIA. (4SQ)

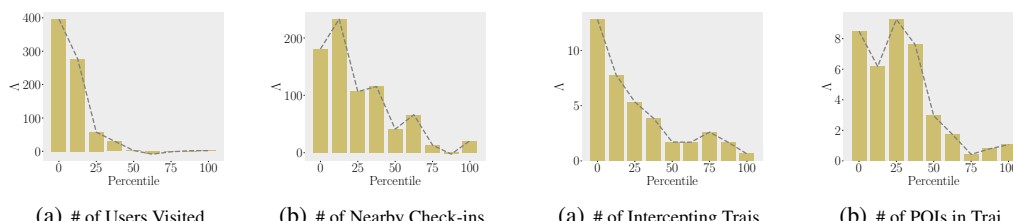

(a)  # of Users Visited     (b)  # of Nearby Check-ins     (a)  # of Intercepting Trajs     (b)  # of POIs in Traj

Figure 4: How location-level aggregate statistics are related to LOCMIA. The locations visited by fewer different users or have fewer surrounding check-ins are more vulnerable to LOCMIA. (4SQ)

Figure 5: How trajectory-level aggregate statistics are related to TRAJMIA. The trajectories with fewer intercepting trajectories or fewer POIs are more vulnerable to TRAJMIA. (4SQ)

In summary, we conclude that the effect of data outliers also exists in privacy attacks against POI recommendations. In the context of POI recommendation, the mobility data outliers could be characterized from the perspectives of user, location, and trajectory. Different attacks in our attack suite might be vulnerable to particular types of data outliers.

### 4.2.3 EXISTING DEFENSES AGAINST OUR ATTACK SUITE

We evaluate existing defenses against our privacy attacks. Due to the limited space, we highlight the key findings in this section and defer experimental details to Appendix E. Specifically, we evaluate two streams of defense mechanisms on proposed attacks, including standard techniques to reduce overfitting (e.g., $l_2$ regularization) and differential privacy (DP) based defenses (e.g., DP-SGD (Abadi et al., 2016)) for provable risk mitigation. Standard techniques are insufficient due to the lack of theoretical guarantees. Moreover, we find that DP-SGD substantially sacrifices the model's utility on the POI recommendation task. The reason is that the training of POI recommendation is highly sensitive to DP noise as the model needs to memorize user-specific patterns from limited user data.

While DP-SGD provides undifferentiated protection for all the mobility data, we argue that only certain sensitive information (e.g., check-ins of a user's home address) needs to be protected in POI recommendation. To this end, we generalize the selective DP method JFT (Shi et al., 2022a) to protect different levels of sensitive information for each attack. Our results show that existing defenses provide a certain level of guarantee in mitigating privacy risks of ML-based POI recommendations. However, there is no such unified defense that can fully tackle the issue with a small utility drop for all attacks, highlighting the need for tailored defenses.

## 5 RELATED WORK

**Mobility Data Privacy**  Mobility data contain rich information that can reveal individual privacy such as user identity. Previous work utilizes side-channel attacks to extract sensitive information about mobility data from LBS, including social relationships (Srivatsa & Hicks, 2012a; Vicente et al., 2011a; Xu et al., 2014), trajectory history (Krumm, 2007b; Golle & Partridge, 2009; Hoh et al., 2006), network packets (Vratonjic et al., 2014; Jiang et al., 2007) and location embeddings Ding et al. (2022). Despite the focus of previous work, deep neural networks (DNN) built on large volumes of mobility data have recently become state-of-the-art backbones for LBS, opening a new surface for privacy attacks. To the best of our knowledge, our work is the first of its kind to investigate the vulnerabilities of DNN models in leaking sensitive information about mobility data using inference attacks.

**Privacy Attacks**  Neural networks could leak details of their training datasets and various types of privacy attacks, such as membership inference attacks (Shokri et al., 2017; Salem et al., 2018; Carlini et al., 2022), training data extraction attacks (Carlini et al., 2019; 2023), and model inversion attacks (Fredrikson et al., 2015b), have been proposed. Our attack suite contains membership inference and data extraction attacks. Existing data extraction and membership inference attacks (Carlini et al., 2019; 2022) are insufficient for POI recommendation models due to the spatio-temporal nature of the data. Our work takes the first step to extracting sensitive location and trajectory patterns from POI recommendation models and solving unique challenges to infer the membership of both user-location pairs and user trajectories. As a final remark, our attacks differ from previous MIAs in mobility data (Pyrgelis et al., 2017; Zhang et al., 2020), which focus on the privacy risks of data aggregation.

## 6 CONCLUSION

In this work, we take the first step to evaluate the privacy risks of the POI recommendation models. In particular, we introduce an attack suite containing data extraction and membership inference attacks to extract and infer sensitive information about location and trajectory in mobility data. We conduct extensive experiments to demonstrate the effectiveness of our attacks. Additionally, we analyze what types of mobility data are vulnerable to the proposed attacks. To mitigate our attacks, we further adapt two mainstream defense mechanisms to the task of POI recommendation. Our results show that there is no single solid defense that can simultaneously defend against proposed attacks. Our findings underscore the urgent need for better privacy-preserving approaches for POI recommendation models.

**Limitations and Future Work**  Moving forward, our future research aims to adapt to the proposed attack to measure the privacy risks of real-world services that utilize private user data in large-scale POI recommendations in more complex settings. For example, label-only attack (Choquette-Choo et al., 2021) is a promising direction for future investigation given it requires less attack knowledge and has unique requirements for data augmentation.

Our attacks have shown that sensitive information can be extracted from POI recommendation models. The existing defense mechanisms are still incapable of simultaneously protecting victim models from all the attacks. This calls for developing better defenses (e.g., machine unlearning (Bourtoule et al., 2021) and fine-tuning models that have been trained on a public dataset (Yu et al., 2023)) contributing to privacy-preserving ML-based POI recommendation systems.

Lastly, our current emphasis is on POI recommendation models. We plan to leave attacks and defense mechanisms to other POI-related tasks, such as POI synthesis (Rao et al., 2020) and POI matching (Ding et al., 2023), in future work.

**Ethics Statement**  Our paper presents a privacy attack suite and evaluates existing state-of-the-art defense mechanisms for POI recommendation models, highlighting the potential privacy vulnerabilities inherent in these POI-focused services. We hope our study fosters future privacy-preserving research for mobility data-based ML models.

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

Table 2: A summary of the threat model.

| Attack | Adversary Objective | Adversary Knowledge |
|--------|---------------------|---------------------|
| LOCEXTRACT | Extract the most frequently visited location $l$ of a target user $u$ | – |
| TRAJEXTRACT | Extract the location sequence of a target user $u$ with length $n$: $x_L = \{l_0, \ldots, l_{n-1}\}$ | Starting location $l_0$ |
| LOCMIA | Infer the membership of a user-location pair $(u,l)$ | Shadow dataset $D_s$ |
| TRAJMIA | Infer the membership of a trajectory sequence $x_T = \{(l_0, t_0), \ldots, (l_n, t_n)\}$ | Shadow dataset $D_s$ |

# A  ATTACK ALGORITHMS

**TRAJEXTRACT**  To start with, we initialize $\beta$ candidate trajectories with the same starting location $l$ and the query time $t$ given a user (lines 1-3). Next, we iteratively extend the candidate trajectories: to extract the $i$-th ($i \geq 1$) locations of $\hat{x}_T^{0:i-1}$ in $\beta$ candidate trajectories, we query the model using $\hat{x}_T^{0:i-1}$ and compute the log perplexity for $\beta \times \mathcal{L}$ possible new trajectories with a length of $i + 1$. We then choose the $\beta$ trajectories with the lowest log perplexity as the new candidate trajectories in the next iteration (line 6). The iterations end until the length of the candidate trajectories reaches $n$. Lastly, we take the location sequences from the final trajectories. Note that both LOCEXTRACT and TRAJEXTRACT need the timestamp $t$ to query the victim model, and we will show the effects of timestamp $t$ in our experiments.

**LiRA**  LiRA consists of three steps: (1) shadow model training, (2) querying victim model and shadow models using $X_{target}$, and (3) conducting a hypothesis test to infer the membership of the $X_{target}$ using the query results. First, shadow model training generates two parameter distributions $\tilde{\mathbb{Q}}_{in}$ and $\tilde{\mathbb{Q}}_{out}$ of the $2N$ shadow models $f_{in}$ and $f_{out}$, respectively. Note that $f_{in}$ ($f_{out}$) corresponds to the shadow model trained with(out) $X_{target}$ and there are $N$ $f_{in}$ and $N$ $f_{in}$, respectively. In LiRA, the goal of $\tilde{\mathbb{Q}}_{in}$ and $\tilde{\mathbb{Q}}_{out}$ is to approximate $\mathbb{Q}_{in}$ and $\mathbb{Q}_{out}$, where $\mathbb{Q}_{in}$ ($\mathbb{Q}_{out}$) is the parameter distribution of $f_\theta$ trained with(out) the $X_{target}$.

After shadow model training, we approximate the values of $\tilde{\mathbb{Q}}_{in}$ ($\tilde{\mathbb{Q}}_{out}$) with the loss (e.g., cross-entropy loss) distributions calculated by querying all shadow models $f_{in}$ and $f_{out}$ with $X_{target}$, i.e., $\tilde{\mathbb{Q}}_{in}(X_{target}, \text{AUX})$ and $\tilde{\mathbb{Q}}_{out}(X_{target}, \text{AUX})$, where AUX is the auxiliary inputs needed to calculate the loss depending on the actual attack.

Finally, we perform a hypothesis test $\Lambda$ to determine the membership of $X_{target}$:

$$\Lambda(f_\theta; X_{target}, \text{AUX}) = \frac{p\big(l_\theta(X_{target}, \text{AUX}) \mid \tilde{\mathbb{Q}}_{in}(X_{target}, \text{AUX})\big)}{p\big(l_\theta(X_{target}, \text{AUX}) \mid \tilde{\mathbb{Q}}_{out}(X_{target}, \text{AUX})\big)}, \tag{1}$$

where $l_\theta(X_{target}, \text{AUX})$ is the loss of querying the victim model $f_\theta$ with $X_{target}$ and $p\big(l_\theta(X_{target}, \text{AUX}) \mid \tilde{\mathbb{Q}}_{in}(X_{target}, \text{AUX})\big)$ is the conditional probability density function of $l_\theta(X_{target}, \text{AUX})$ given $\tilde{\mathbb{Q}}_{in}(X_{target}, \text{AUX})$. The likelihood ratio $\Lambda$ can be used to determine if we should reject the hypothesis that $f_\theta$ is trained on $X_{target}$. In practice, we need both $\tilde{\mathbb{Q}}_{in}$ and $\tilde{\mathbb{Q}}_{out}$ follow Gaussian distributions so that Eq. 1 has a closed form solution. Thus, we use the logit scaling method (Carlini et al., 2022) (i.e., $\phi = \log(\frac{p}{p-1})$ where $p = f(X_{target})_y$) in the calculation of $l_\theta$, $\tilde{\mathbb{Q}}_{in}$, and $\tilde{\mathbb{Q}}_{in}$.

LiRA originally trained $2N$ shadow models for each target example. However, this approach suffers from computational inefficiency when the number of target examples is large. To address this issue,

we employ the parallelized approach described in (Carlini et al., 2022), which reuses the same set of $2N$ shadow models for inferring the membership of multiple $X_{target}$.

---

**Algorithm 1** Common Location Extraction Attack

---

**Input:** Victim model: $f_\theta$, target user: $u$, query budget: $q$, query timestamp: $t$, output size: $k$
**Output:** Top-$k$ predictions: $[\hat{l}_{top1}, \ldots, \hat{l}_{topk}]$
1: logits $\leftarrow \{\}$
2: **for** $q$ times **do**
3:      $l \leftarrow$ RANDOMSAMPLE$(\mathcal{L})$ ▷ *Randomly generate a location from the location space*
4:      logits $\cup\, f_\theta\big(u, \{(l,t)\}\big)$
5: **end for**
6: logits$_{\text{agg}}$ = AGGREGATE(logits) ▷ *Aggregate confidence for all locations*
7: **return** $\hat{l}_{top1}, \ldots, \hat{l}_{topk} \leftarrow$ ARGMAX$_k$(logits$_{\text{agg}}$)

---

---

**Algorithm 2** Training Trajectory Extraction Attack

---

**Input:** Victim model: $f_\theta$, target user: $u$, starting location: $l_0$, target extraction length: $n$, query timestamp: $t$, beam width: $\beta$
**Output:** Top-$\beta$ possible extraction results: $\hat{x}_{L_0}^{0:n}, \ldots, \hat{x}_{L_\beta}^{0:n}$
1: **for** $b \leftarrow 0$ to $\beta - 1$ **do**
2:      $\hat{x}_{T_b}^{0:0} \leftarrow (u, (l_0, t))$ ▷ *Initialize the beam with $l_0$ and $t$*
3: **end for**
4: **for** $i \leftarrow 1$ to $n - 1$ **do**
5:      **for** $\hat{x}_T^{0:i-1}$ in $\{\hat{x}_{T_0}^{0:i-1}, \ldots, \hat{x}_{T_\beta}^{0:i-1}\}$ **do**
6:          $\{\hat{x}_{T_0}^{0:i}, \ldots, \hat{x}_{T_\beta}^{0:i}\} \leftarrow$ UPDATEBEAM$_\beta(f_\theta(u, \hat{x}_T^{0:i-1}))$ ▷ *Update the beam by keeping $\beta$ trajectory with the smallest PPL from the query output and current beam*
7:      **end for**
8: **end for**
9: $\hat{x}_{L_0}^{0:n-1}, \ldots, \hat{x}_{L_\beta}^{0:n-1} \leftarrow$ GETLOC$(\hat{x}_{T_0}^{0:n-1}, \ldots, \hat{x}_{T_\beta}^{0:n-1})$ ▷ *Take the location sequence from $\hat{x}_T^{0:n-1}$ as result $\hat{x}_L^{0:n-1}$*
10: **return** $\hat{x}_{L_0}^{0:n-1}, \ldots, \hat{x}_{L_\beta}^{0:n-1}$

---

---

**Algorithm 3** SPATEMQUERY: Spatial-Temporal Model Query Algorithm for LOCMIA

---

**Input:** Target model: $f_{target}$, number of query timestamps: $n_t$, number of query locations: $n_l$, target example: $X_{target}$
**Output:** Membership confidence score: $conf$
1: $u, l \leftarrow X_{target}$
2: $conf_{all} \leftarrow \{\}$
3: **for** $i \leftarrow 0$ to $n_t - 1$ **do**
4:      $conf_t \leftarrow \{\}$
5:      **for** $j \leftarrow 0$ to $n_l - 1$ **do**
6:          $t_i \leftarrow i/n_t$
7:          $l_j \leftarrow$ RANDOMSAMPLE$(\mathcal{L})$
8:          $conf_t \leftarrow conf_t \cup f_{target}(u, (l_j, t_i))$ ▷ *Query the model with random location and a synthetic timestamp*
9:      **end for**
10:     $conf_{all} \leftarrow conf_{all} \cup \text{mean}(conf_t)$ ▷ *Calculate average confidence from all queries for this timestamp*
11: **end for**
12: **return** $conf \leftarrow \max(conf_{all})$ ▷ *Take the confidence scores with largest confidence at position $l$ as output*

---

---

**Algorithm 4** Membership Inference Attack

---

Below, we demonstrate our location-level MIA and trajectory-level MIA algorithms. The lines marked in red are specific to LOCMIA, while the lines marked in purple are specific to TRAJMIA. Both attacks share the remaining lines.

**Input:** Victim model: $f_\theta$, shadow data: $D_s$, number of shadow models: $N$, extraction target: $X_{target}$, shadow timestamp: $t_s$, number of query timestamps: $n_t$, number of query locations: $n_l$

**Output:** The likelihood ratio to determine if we should reject the hypothesis that $X_{target}$ is a member of $f_\theta$: $\Lambda$

1: $conf_{in}, conf_{out} \leftarrow \{\},\{\}$
2: $X_S \leftarrow \text{RANDOMSAMPLE}(\{X_S : X_{target} \in X_S\})$ ▷ *Sample a location sequence that includes* $X_{target}$
3: $X_S \leftarrow X_{target}$
4: **for** $i \leftarrow 0$ to $N$ **do**
5: $\quad D_{in} \leftarrow \text{RANDOMSAMPLE}(D_s) \cup X_S$
6: $\quad D_{out} \leftarrow \text{RANDOMSAMPLE}(D_s) \backslash X_S$
7: $\quad f_{in}, f_{out} \leftarrow \text{TRAIN}(D_{in}), \text{TRAIN}(D_{out})$ ▷ *Train $f_{in}$ and $f_{out}$*
8: $\quad conf_{in} \leftarrow conf_{in} \cup \phi(\text{SPATEMQUERY}(f_{in}, n_t, n_l, X_S))$
9: $\quad conf_{out} \leftarrow conf_{out} \cup \phi(\text{SPATEMQUERY}(f_{out}, n_t, n_l, X_S))$
10: $\quad conf_{in} \leftarrow conf_{in} \cup \phi\big(\text{mean}(\{f_{in}(X_S)^{0:0}, \ldots, f_{in}(X_S)^{0:n-1}\})\big)$
11: $\quad conf_{out} \leftarrow conf_{out} \cup \phi\big(\text{mean}(\{f_{out}(X_S)^{0:0}, \ldots, f_{out}(X_S)^{0:n-1}\})\big)$
12: **end for**
13: $\mu_{\text{in}}, \mu_{\text{out}} \leftarrow \text{mean}(conf_{\text{in}}), \text{mean}(conf_{\text{out}})$
14: $\sigma_{\text{in}}^2, \sigma_{\text{out}}^2 \leftarrow \text{var}(conf_{\text{in}}), \text{var}(conf_{\text{out}})$
15: $conf_{\text{obs}} \leftarrow \phi(\text{SPATEMQUERY}(f_\theta, n_t, n_l, X_S))$
16: $conf_{\text{obs}} \leftarrow \phi\big(\text{mean}(\{f_\theta(X_S)^{0:0}, \ldots, f_\theta(X_S)^{0:n-1}\})\big)$
17: **return** $\Lambda = \frac{p(conf_{\text{obs}}|\mathcal{N}(\mu_{\text{in}}, \sigma_{\text{in}}^2))}{p(conf_{\text{obs}}|\mathcal{N}(\mu_{\text{out}}, \sigma_{\text{out}}^2))}$ ▷ *Hypothesis test*

Table 3: Statistics of POI Recommendation Datasets.

| | #POIs | #Check-ins | #Users | #Trajectories | Avg. Len. |
|---|---|---|---|---|---|
| 4SQ | 4,556 | 63,648 | 1,070 | 17,700 | 3.63 |
| GOWALLA | 2,559 | 32,633 | 1,419 | 7,256 | 4.46 |

## B    DETAILED EXPERIMENTAL SETUP

### B.1    DATASETS

We conduct experiments on two POI recommendation benchmark datasets – FourSquare (4SQ) (Yang et al., 2014) and GOWALLA (Cho et al., 2011) datasets. Following the literature (Yang et al., 2022; Kong & Wu, 2018), we use the check-ins collected in NYC for both sources. The 4SQ dataset consists of 76,481 check-ins during ten months (from April 12, 2012, to February 16, 2013). The GOWALLA dataset comprises 35,674 check-ins collected over a duration of 20 months (from February 2009 to October 2010). In both datasets, a check-in record can be represented as [user ID, check-in time, latitude, longitude, location ID].

### B.2    DATA PREPROCESSING

We preprocess each dataset following the literature (Yang et al., 2022): (1) We first filter out unpopular POIs and users that appear less than ten times to reduce noises introduced by uncommon check-ins. (2) To construct trajectories of different users in a daily manner, the entire check-in sequence of each user is divided into trajectories with 24-hour intervals. Then, we filter out the trajectories with only a single check-in. (3) We further normalize the timestamp (from 0:00 AM to 11:59 PM) in each check-in record into $[0, 1]$. After the aforementioned steps, the key statistics of the 4SQ and GOWALLA datasets are shown in Table 3. (4) Lastly, we split the datasets into the training, validation, and test sets using the ratio of 8:1:1.

**Victim Model Training Settings:** We use the official implementation of GETNEXT[2] and LSTPM[3] to train victim models. In particular, we train each model with a batch size 32 for 200 epochs by default. We use five random seeds in all experiments and report the average results.

## B.3 ATTACK SETTINGS

**LOCEXTRACT** Given a target user $u$, we extract the most visited location $l_{top1}$ from the victim model $f_\theta$ with a query number $q = 50$. We set the query timestamp $t = 0.5$ (i.e., the middle of the day) by default, and we will present how the change of the query timestamp affects the attack performance in the ablation study.

**TRAJEXTRACT** In this attack, we experiment with $n = 4$ by default, though the attacker can potentially extract location (sub-)sequences with arbitrary length. We set the beam size $\beta = 50$ in the beam search to query the victim model and update candidate trajectories. For each query, we also have the default query timestamp $t = 0.5$.

**LOCMIA** In our experiments, since we randomly sample 80% of trajectories as the training dataset $D_{tr}$ to build a victim model for MIA, we treat the remaining 20% data as non-members. For each target user $u$ and the POI location $l$ pair, we generate $N = 64$ synthesis trajectories using TRAJSYNTHESIS with the query timestamp $t_s = 0.5$. With the synthesis trajectories, we can also have 64 in-models ($f_{in}$) and 64 out-models ($f_{out}$). We also set $n_t = 10$ and $n_l = 10$. For evaluation, we conduct a hypothesis test on a balanced number of members and non-members.

**TRAJMIA** We extract the membership information of some trajectory sequences with arbitrary lengths from the victim model. We also build $N = 64$ in-models ($f_{in}$) and $N = 64$ out-models ($f_{out}$) for a target trajectory sequence. For evaluation, we conduct a hypothesis test on a balanced number of members and non-members.

## C MORE DETAILS OF ANALYZING FACTORS IN MOBILITY DATA THAT MAKE IT VULNERABLE TO THE ATTACKS

### C.1 HOW TO SELECT AGGREGATE STATISTICS

This section outlines the basic principles and details for selecting representative aggregate statistics for analysis. For user-level aggregate statistics, we target the basic statistical information quantifying properties of locations and trajectories of a user. For location-level and trajectory-level aggregate statistics, we study their users, "neighboring" check-ins and trajectories, and the check-in time information. In summary, we select the following aggregate statistics:

- User-level aggregate statistics:
    1. Total number of check-ins;
    2. Number of unique visited POIs;
    3. Number of trajectories;
    4. Average trajectory length;
- Location-level aggregate statistics:
    1. Number of users who have visited this POI;
    2. Number of check-ins surrounding ($\leq$ 1km) this POI;
    3. Number of trajectories sharing this POI;
    4. Average time in a day for the visits to the POI;
- Trajectory-level aggregate statistics:
    1. Number of users who have the same trajectories;
    2. Number of check-ins surrounding ($\leq$ 1km) all POI in the trajectory;
    3. Number of intercepting trajectories;
    4. Average check-in time of the trajectory.

---

[2]https://github.com/songyangme/GETNext
[3]https://github.com/NLPWM-WHU/LSTPM

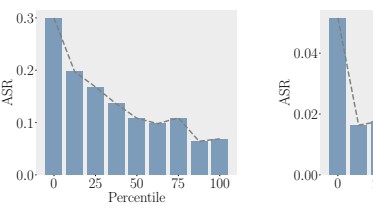

(a) # Unique POIs (4SQ)    (b) # Unique POIs (GOWALLA)

Figure 6: How user-level aggregate statistics are related to TRAJEXTRACT. The users who have fewer unique POIs are more vulnerable to TRAJEXTRACT.

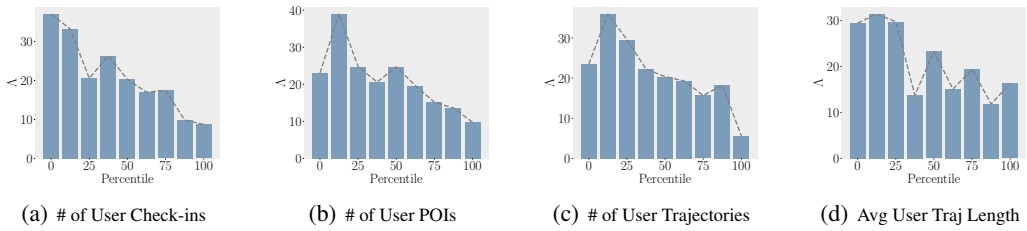

(a) # of User Check-ins    (b) # of User POIs    (c) # of User Trajectories    (d) Avg User Traj Length

Figure 7: How user-level aggregate statistics are related to TRAJMIA. The users with fewer total check-ins, fewer unique POIs, and fewer or shorter trajectories are more vulnerable to TRAJMIA. (GOWALLA)

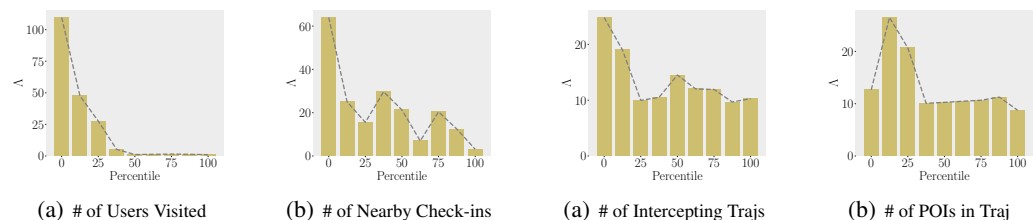

(a) # of Users Visited    (b) # of Nearby Check-ins    (a) # of Intercepting Trajs    (b) # of POIs in Traj

Figure 8: How location-level aggregate statistics are related to LOCMIA. The locations shared by fewer users or have fewer surrounding check-ins are more vulnerable to LOCMIA. (GOWALLA)

Figure 9: How trajectory-level aggregate statistics are related to TRAJMIA. The trajectories with fewer intercepting trajectories or fewer POIs in the trajectory are more vulnerable to TRAJMIA. (GOWALLA)

## D   THE IMPACT OF TRAINING AND ATTACK PARAMETERS

In this section, we analyze how training parameters (Sec. D.1) and attack parameters affect the attack performance of data extraction attack (Sec. D.2) and membership inference attack (Sec. D.3).

### D.1   THE IMPACT OF TRAINING PARAMETERS

Our primary objective of this analysis is to understand whether the occurrence of overfitting, commonly associated with excessive training epochs, leads to heightened information leakage based on our proposed attacks. By showcasing the ASR at various model training stages, we aim to gain insights into the relationship between overfitting and ASR.

Based on the results presented in Figure 10, we observe that LOCEXTRACT achieves the best performance when the model is in the convergence stage. We speculate that continuing training beyond convergence leads to overfitting, causing a loss of generalization. Specifically, when the model is overfitted, it tends to assign higher confidence to the training data while disregarding the general rules present in the dataset. In contrast, our attack employs random queries to extract the general rules learned by the model from the dataset, resulting in better performance when applied to the optimally fitted model.

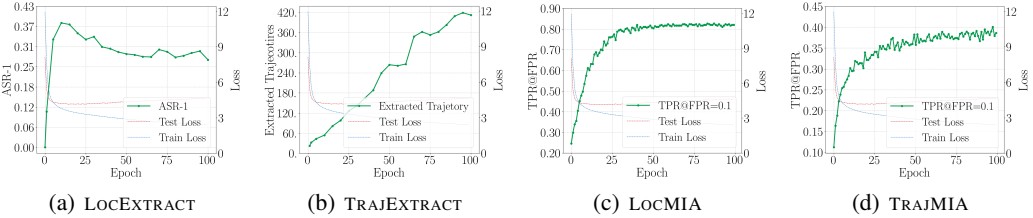

(a) LOCEXTRACT  (b) TRAJEXTRACT  (c) LOCMIA  (d) TRAJMIA

Figure 10: The impact of the model generalization on the performance of four privacy attacks.

The attack performance of TRAJEXTRACTimproves as the training process progresses, which can be attributed to the model becoming increasingly overfitted. The overfit model is more likely to output the exact training trajectory and generates more precise training trajectories than the best-fitted model when given the same number of queries. Similarly, the results of our membership inference attacks reveal a trend of attack performance consistently improving with the progression of the training process. This observation aligns with our expectations, as when the model undergoes more training iterations, the effects of training data are more emphasized. Consequently, the distribution of query results in our attack on the seen training data diverging further from the distribution derived from the unseen data. This growing disparity between the two distributions facilitates the membership inference task, particularly on overfitted models. The analysis of these three attacks indicates a consistent trend, highlighting the increased risk of privacy leakage due to overfitting with respect to the original training data.

## D.2   ABLATION STUDY ON OUR DATA EXTRACTION ATTACKS

**Data extraction attacks are effective given a limited number of queries**   In a realistic attack scenario, the adversary may encounter query limitations imposed by the victim model, allowing the adversary to query the model for only a certain number of queries. Figure 11 illustrates that our data extraction attacks are effective given a limited number of queries. For example, as shown in Figure 11(a), a mere $q = 50$ query is sufficient for the adversary to achieve a high ASR and infer a user's frequently visited location. In terms of TRAJEXTRACT attack (Figure 11(b)), the adversary can opt for a small beam width of $\beta = 10$, requiring only 1000 queries to extract a trajectory of length $n = 4$. This practicality of our data extraction attack holds true even when the query limit is very small.

**Appropriate query timestamp improves the effectiveness of data extraction attacks**   POI recommendation models rely on temporal information to make accurate location predictions. However, obtaining the same timestamps as training for attack can be challenging and is an unrealistic assumption. Therefore, in our data extraction attack setup, we set the query timestamp to $t = 0.5$ (i.e., the middle of the day).

To analyze the effect of how different query timestamps affect data extraction attack performance, we conduct extraction attacks and vary different timestamps that represent various sections within a 24-hour window in the experiments. The results, illustrated in Figure 12, indicate that utilizing timestamps corresponding to common check-in times, such as the middle of the day or late afternoon, yields better attack outcomes. This finding aligns with the rationale that users are more likely to engage in check-ins during the daytime or after work hours. Consequently, locations associated with common check-in times exhibit a higher likelihood of being connected to the most frequently visited locations.

**Soft voting improves LOCEXTRACT**   For LOCEXTRACT, we have the option to employ either hard voting or soft voting to determine the most frequently occurring location. Hard-voting ensembles make predictions based on a majority vote for each query while soft-voting ensembles consider the average predicted probabilities and select the top-k locations with the highest probabilities. From the experimental results depicted in Figure 13, we observe that there is not a substantial difference in ASR-1 when using hard voting or soft voting. However, employing soft voting yields better ASR-3 and ASR-5 results.

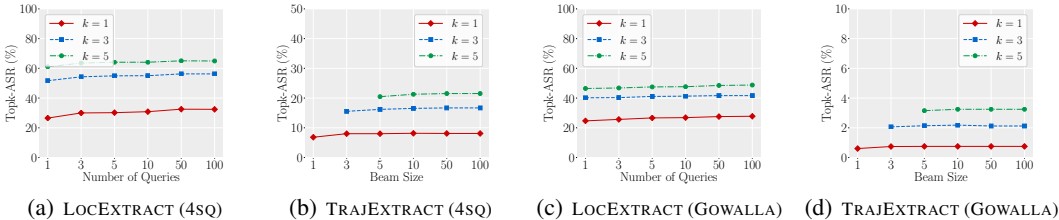

(a) LOCEXTRACT (4SQ)  (b) TRAJEXTRACT (4SQ)  (c) LOCEXTRACT (GOWALLA)  (d) TRAJEXTRACT (GOWALLA)

Figure 11: Our LOCEXTRACT is effective with a small number of queries and TRAJEXTRACT is effective with a small beam size (i.e., both attacks are effective within a small query budget).

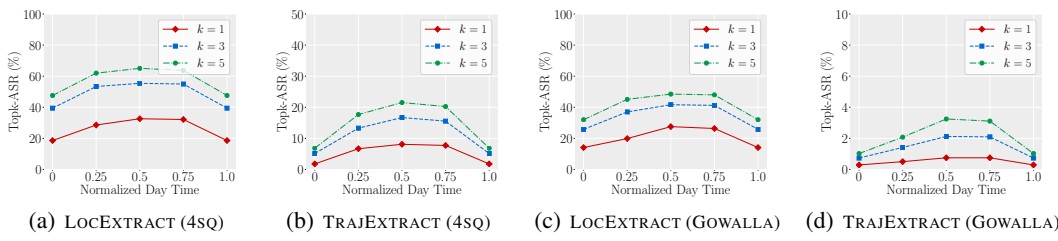

(a) LOCEXTRACT (4SQ)  (b) TRAJEXTRACT (4SQ)  (c) LOCEXTRACT (GOWALLA)  (d) TRAJEXTRACT (GOWALLA)

Figure 12: The optimal query timestamp can significantly improve the performance of LOCEXTRACT and TRAJEXTRACT.

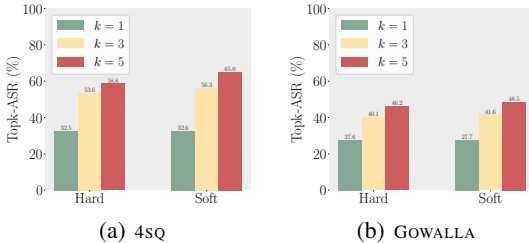

(a) 4SQ  (b) GOWALLA

Figure 13: Comparing soft voting with hard voting for logits aggregation in LOCEXTRACT. Soft voting has larger improvements over hard voting as $k$ increases.

**Location sequences of shorter trajectories are more vulnerable to TRAJEXTRACT**  For TRAJEXTRACT, we conduct an ablation study to extract trajectories of varying lengths $n$. The results, illustrated in Figure 14, indicate that the attack achieves higher ASR on shorter trajectories than longer ones. This observation can be attributed to our assumption that the attacker possesses prior knowledge of a starting location. As the prediction moves further away from the starting location, its influence on subsequent locations becomes weaker. Consequently, predicting locations farther from the starting point becomes more challenging, decreasing the attack's success rate for longer trajectories. Moreover, the extraction of long trajectories presents additional difficulties. With each step, the probability of obtaining an incorrect location prediction increases, amplifying the challenges the attack algorithm faces.

### D.3  ABLATION STUDY ON OUR MEMBERSHIP INFERENCE ATTACKS

**A larger number of shadow models improves the effectiveness of MIAs**  As mentioned in Carlini et al. (2022), it has been observed that the attack's performance of LiRA tends to improve as the number of shadow models increases. Consistently, our attacks also follow this pattern, as depicted in Figure 15. Both location-level MIA and trajectory-level MIA show enhanced performance as we incorporate more shadow models. This improvement is because an increased number of shadow

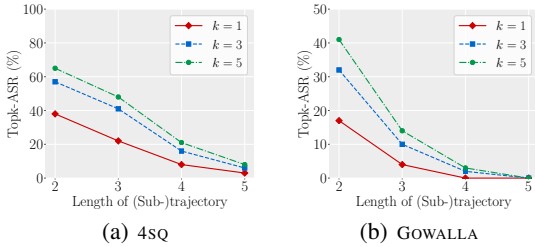

Figure 14: The location sequences of shorter (sub-)trajectories are more vulnerable to TRAJEXTRACT.

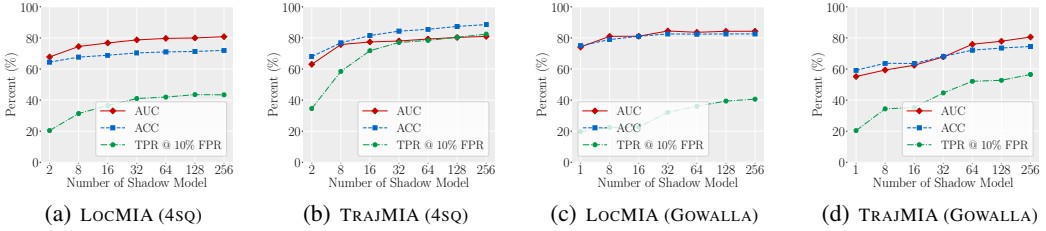

Figure 15: The attack performance of LOCMIA and TRAJMIA significantly improves as the number of shadow model increases.

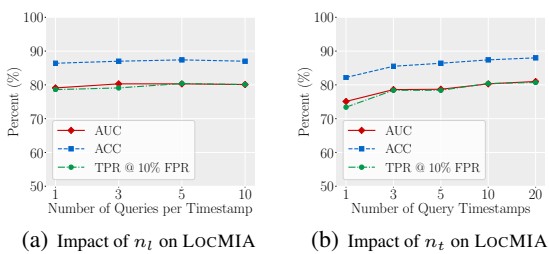

Figure 16: Both the number of query timestamps $n_t$ and the number of query locations $n_l$ affect the attack performance of LOCMIA. We use the default $n_t = 10$ and $n_l = 10$ in (a) and (b), respectively. Given a few queries, our LOCMIA remains effective.

models allows for a better approximation of the distributions for $f_{in}$ and $f_{out}$, thereby simulating the victim model more accurately.

**LOCMIA is effective given a limited number of queries** Since our LOCMIA involves multiple queries to explore locations preceding the target location, as well as the corresponding timestamps, it is essential to consider potential limitations on the number of queries in real-world scenarios. Thus, we conduct experiments to investigate the impact of query limits on LOCMIA. The results, depicted in Figure 16(a), indicate that our attack remains effective even with a limited number of queries for different location choices. The further increase in query locations would not significantly improve attack results.

We also conduct experiments with different settings for the number of query timestamps, denoted as $n_t$. The rationale behind this step is that the adversary does not possess information about the real timestamp used to train the victim model. To simulate the effect of selecting the correct timestamp, we perform experiments with varying timestamps to identify the timestamp that yielded the highest confidence score for the targeted location. Based on empirical observations from our experiment on the 4SQ dataset (see Figure 16(b)), increasing the number of query timestamps tends to yield better overall results in practice.

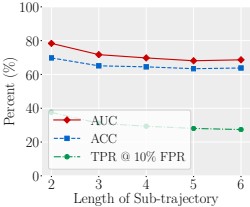

Figure 17: The longer trajectories are less vulnerable to TRAJMIA.

**TRAJMIA is less effective as the length of target trajectory increases**  From Figure 17, we note that the attack performance drops as target trajectories are longer. This decline happens because all trajectory query scores influence the attack. Lengthier trajectories introduce increased randomness for the query, affecting the outcome of TRAJMIA.

Table 4: Sensitive information defined in each attack.

| Attack | Sensitive Information |
|---|---|
| LOCEXTRACT | Most common location of each user |
| TRAJEXTRACT | Each location sequence/sub-sequence $(x_L)$ |
| LOCMIA | Each user-location pair $(u, l)$ |
| TRAJMIA | Each trajectory sequence/sub-sequence $(x_T)$ |

## E  DEFENSE

In this section, we generalize and evaluate existing defenses against our privacy attacks. In E.1, we illustrate the metrics to measure the defense performance. In E.2, we describe the defense mechanisms in detail. In E.3, we compare different defenses and analyze the numerical results with detailed explanations.

### E.1  DEFENSE METRICS

Our inference attacks extract different sensitive information about the training dataset from the victim model, as summarized in Table 4. To this end, we evaluate defense mechanisms in terms of their performance in preventing each attack from stealing the corresponding sensitive information. Specifically, we measure their defense performance on protecting *all the sensitive information* and *a targeted subset of sensitive information* for each attack, respectively. Here, we define the targeted subset of sensitive information as the mobility data a defender wants to protect in practice (e.g., some selected user-location pairs in LOCMIA). We introduce this metric because not all mobility data are sensitive or equally important. Take LOCMIA as an example: Since the utility of POI recommendation is highly related to the model's memorization of user-location pairs, a user may want the model to recognize most of the POIs in his trajectory history while hiding those that are very likely to leak his personal identity (e.g., home). In other words, not all the mobility data need to be protected and it's more important to evaluate how defense mechanisms perform on the targeted subset of sensitive information.

To this end, we jointly measure the defense performance in protecting all the sensitive information and the targeted subset of sensitive information for each attack. Based on different attack objectives, we construct a different targeted subset of sensitive information for measurement by randomly sampling a portion of (e.g., 30%) the most common locations in LOCEXTRACT, location sequences in TRAJEXTRACT, user-location pairs in LOCMIA and trajectory sequences in TRAJMIA. It is noted that we randomly sample 30% of sensitive information in each attack to construct the targeted subset for the ease of experiments. In practice, the defender may have more personalized choices based on user-specific requirements, which we leave as future work.

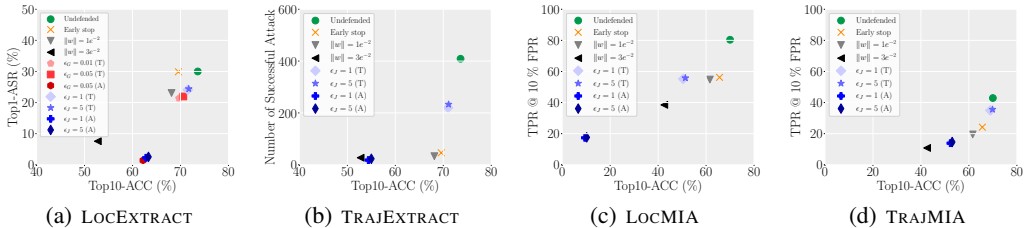

Figure 18: Defense performance on protecting all corresponding sensitive information for each attack.

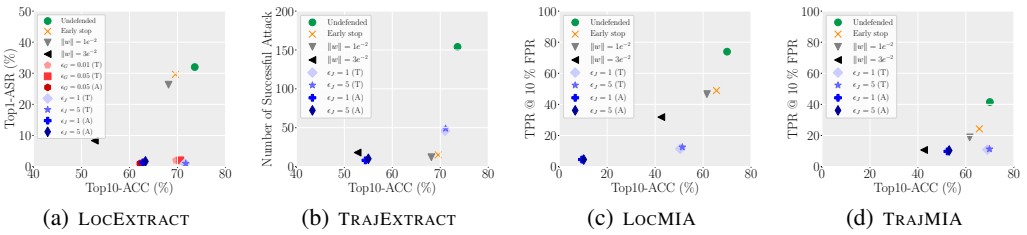

Figure 19: Defense performance on protecting the targeted subset of sensitive information for each attack.

### E.2 DEFENSE TECHNIQUES

We now evaluate two streams of defense mechanisms on proposed attacks, including standard ML techniques for reducing memorization and differential privacy (DP) based defenses for provable risk mitigation. For standard techniques, we apply $L_2$ norm regularization and early stopping during training to reduce the victim model's memorization to some degree. However, due to the lack of theoretical guarantees, the victim model still leaks certain levels of sensitive information when these techniques are applied. To fill this gap, differential privacy (Dwork et al., 2014) is also used to defend against our attacks, which can theoretically limit the impact of any single data point on the final outcomes, according to Definition 1.

We first experiment with DP-SGD (Abadi et al., 2016), the most representative DP-based defense, to train differentially-private POI recommendation models. The key idea of DP-SGD is to add Gaussian noises $\mathcal{N}(0, \sigma^2 C^2 I)$ to the clipped gradients $g$ of the model during its training process. Here, $C$ indicates a clipping threshold that bounds the sensitivity of $g$ by ensuring $\|g\| \leq C$. To achieve $(\epsilon, \delta)$-DP, we have $\sigma = \sqrt{2 \ln \frac{1.25}{\delta}}/\epsilon$. Despite that DP-SGD achieves promising results on some language tasks, we find that it can substantially sacrifice the model's utility on the POI recommendation task. Specifically, the top-10 accuracy is only 4.97% when the mechanism satisfies $(5, 0.001)$-DP, while the original top-10 accuracy without DP is 71%. The reason is that POI recommendation aims to make user-level predictions within a large input (e.g., > 1k users) and output (e.g., a large number of possible POIs) space. For different users, even the same location sequence may lead to different results. In other words, the model needs to learn user-specific patterns from very limited user-level training data. As a result, the training is quite sensitive to the noises introduced by DP-SGD, making it not applicable to POI recommendations.

We note that DP-SGD provides undifferentiated protection for all the mobility data, which causes such poor utility. However, for each attack, only the sensitive information needs to be protected. Moreover, a defender may only care about whether the targeted subset of sensitive information is protected or not. To this end, we use the notion of selective DP (Shi et al., 2022b) to relax DP and improve the model's utility-privacy trade-offs. Specifically, we apply the state-of-the-art selective DP method JFT (Shi et al., 2022a) to protect different levels of sensitive information for each attack. The key idea of JFT is to adopt a two-phase training process: in the phase-I training, JFT redacts the sensitive information in the training dataset and optimizes the model with a standard optimizer; in the phase-II training, JFT applies DP-SGD to finetune the model on the original dataset in a

privacy-preserving manner. Because of the phase-I training, we observe that the model's utility is significantly promoted.

In addition to JFT, we also apply Geo-Indistinguishability (Geo-Ind) (Andrés et al., 2013) to secure location data in LOCEXTRACT. Geo-Ind replaces each target POI with a nearby location based on the Laplacian mechanism. In this way, a redacted location tells nothing about the original POI and its neighboring locations ($\leq r$) with $\epsilon r$-privacy guarantees, as shown in 2. We note that Geo-Ind is only applicable to (most common) locations in LOCEXTRACT (but not LOCMIA) as it requires modifying the training data and is incompatible with the notion of membership. Next, we will illustrate the defense performance of regularization, early stopping, JFT, and Geo-Ind on proposed attacks.

### E.3 DEFENSE SETUP AND RESULTS

**Setup** GETNext models trained on the 4sq dataset are used for experiments. For $L_2$ regularization, we use weight decay $\|w\| = 1e^{-2}$ and $3e^{-2}$. For early stopping, we stop the training after 5 epochs. For JFT, we mask sensitive information that needs to be protected in phase-I. Then in phase-II, we use DP-SGD (Abadi et al., 2016) with different $\epsilon_J$ (1 and 5) to finetune the model. The $C$ and $\delta$ are set to 10 and $1e^{-3}$. For Geo-Ind against LOCEXTRACT, we apply different $\epsilon_G$ (0.01 and 0.05) to replace each sensitive POI with its nearby location such that the original POI is indistinguishable from any location within $r = 400$ meters. Since both JFT and Geo-Ind can be used to protect different amounts of sensitive information, we either protect nearly all the sensitive information or only the targeted subset of sensitive information for each attack, denoted by suffixes (A) and (T).

**LOCEXTRACT** Figure 18(a) shows the defense results on protecting all the sensitive information for LOCEXTRACT. From the figure, we observe that DP-based defenses achieve better performance than standard techniques. Both JFT (A) and Geo-Ind (A) reduce ASR from 30% to 1% with only a 10% drop in utility. The reason is that these methods allow the defender to selectively protect common locations only. Besides, when protecting the same amount of sensitive information, JFT achieves slightly better accuracy than Geo-Ind because it involves phase-II training to further optimize the model. Moreover, although the ASR is still high for JFT (T) and Geo-Ind (T) in Figure 18(a), we notice that they can substantially reduce the attack performance on the targeted subset of sensitive information, as shown in Figure 19(a). This allows a defender to protect the targeted subset with negligible utility drop. Figure 20 further shows that Geo-Ind can predict nearby locations of a protected POI as prediction results to maintain its usage.

**TRAJEXTRACT** Figures 18(b) and 19(b) show that all the defenses can well protect location (sub-)sequences from being extracted by TRAJEXTRACT. This is because sequence-level extraction is a challenging task that pretty much relies on memorization.

**LOCMIA** Figures 18(c) and 19(c) show that none of the existing defenses can be used to protect user-location membership information. While JFT (A) reduces the TPR@10%FPR to less than 20%, it significantly sacrifices the model's utility. The reason is that the defender needs to redact a large number of user-location pairs so as to protect them. As a result, the model may learn from wrong sequential information in the phase-I training, leading to a large utility drop. Even for protecting the targeted subset with 30% of total user-location pairs only, there's still a 20% drop in utility.

**TRAJMIA** Figures 18(d) and 19(d) show the utility-privacy trade-off of different defenses against TRAJMIA We notice that JFT (T) can effectively mitigate the MIA on the targeted subset of trajectory sequences with a small degradation in accuracy. However, the utility drop is still large if a defender aims to protect the membership information of all trajectory sequences.

**Summary** Existing defenses provide a certain level of guarantee in mitigating the privacy risks of ML-based POI recommendations. However, it is still challenging to remove the victim model's vulnerabilities within a reasonable utility drop. This is because existing POI recommendation models rely heavily on memorizing user-specific trajectory patterns that lack sufficient semantic information (e.g., compared to text in NLP). As a result, defense mechanisms such as DP-SGD can easily compromise the utility due to their noises on the gradients. Moreover, defenses such as JFT are not general for all inference attacks since each attack targets different sensitive information. To this end, our evaluation calls for more advanced mechanisms to defend against our attacks.

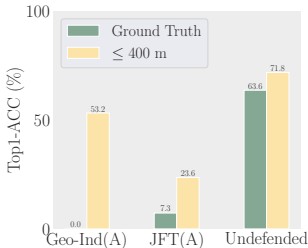

Figure 20: For LOCEXTRACT, when ground truths are the most common locations, Geo-Ind (A) can predict a nearby location ($\leq 400$ m) of each protected location as the next POI with higher accuracy than JFT (A). The reason is that Geo-Ind applies the Laplacian mechanism to replace each protected POI with its nearby location.

### E.4 ADDITIONAL DEFINITIONS IN DEFENSE

**Definition 1 (($\epsilon, \delta$)-DP)** *A randomized mechanism $\mathcal{A}$ satisfies ($\epsilon, \delta$)-DP if and only if for any two adjacent datasets $D$ and $D'$, we have:*

$$\forall \mathcal{O} \in Range(\mathcal{A}) : Pr[\mathcal{A}(D) \in \mathcal{O}] \leq e^\epsilon Pr[\mathcal{A}(D') \in \mathcal{O}] + \delta$$

*where $Range(A)$ indicates the set of all possible outcomes of mechanism $\mathcal{A}$ and $\delta$ indicates the possibility that plain $\epsilon$-differential privacy is broken.*

**Definition 2 (geo-indistinguishability)** *A mechanism $\mathcal{A}$ satisfies $\epsilon$-geo-indistinguishability iff for all $l$ and $l'$ , we have:*

$$d_\mathcal{P}(\mathcal{A}(l), \mathcal{A}(l')) \leq \epsilon d(l, l')$$

*where $d$ denotes the Euclidean metric and $d_\mathcal{P}$ denotes the distance between two output distributions. Enjoying $\epsilon r$-privacy within $r$ indicates that for any $l$ and $l'$ such that $d(l, l') \leq r$, mechanism $\mathcal{A}$ satisfies $\epsilon$-geo-indistinguishability.*

## F   MORE RELATED WORK ON DEFENSES AGAINST PRIVACY ATTACKS

**Defenses against privacy attacks on mobility data**   As discussed in Section 5, there have been various studies on stealing sensitive information from mobility data. Consequently, researchers have also explored various approaches to safeguard the privacy of mobility data, including K-Anonymity (Gedik & Liu, 2005; Gruteser & Grunwald, 2003), which aims to generalize sensitive locations by grouping them with other locations, Location Spoofing (Bordenabe et al., 2014; Hara et al., 2016), which involves sending both real and dummy locations to deceive adversaries, Geo-indistinguishability (Yan et al., 2022; Andrés et al., 2013), and local differential privacy (LDP) (Xu et al., 2023; Bao et al., 2021). However, these prior defense mechanisms primarily focus on data aggregation and release processes and can not be directly used in the context of POI recommendation. In contrast, our work is the first to concentrate on protecting privacy breaches originating from deep learning models such as POI recommendation models.

**Defenses against privacy attacks on deep neural networks**   There are also multiple works on protecting the privacy of deep learning models, with some notable examples including regularization and early stopping, which are commonly employed techniques to mitigate overfitting (Goodfellow et al., 2016). Another approach is differentially private stochastic gradient descent (DP-SGD) (Abadi et al., 2016), which achieves differential privacy by introducing noise during the gradient descent process while training the model. Additionally, selective differential privacy (S-DP) has been proposed to safeguard the privacy of specific subsets of a dataset with a guarantee of differential privacy (Shi et al., 2022b;a). However, these methods have primarily been tested on image or language-related models and require customization to fit into the usage of POI recommendation models. In our work, we focus on adapting these defense mechanisms to POI recommendation models by developing privacy definitions that are specifically tailored to the attacks we propose. In addition, we follow the concept of selective DP (Shi et al., 2022a) to relax the original DP and selectively protect the sensitive information (e.g., most common locations) in mobility data.

