# OpenReview forum: "Where have you been? A Study of Privacy Risk for Point-of-Interest Recommendation"
_ICLR.cc/2024/Conference — Submitted to ICLR 2024_

### Official Review · Reviewer_KUtq · 2023-10-27

**Soundness:** 3 good
**Presentation:** 3 good
**Contribution:** 3 good
**Rating:** 5
**Confidence:** 4

**Summary:**

The paper emphasizes the importance of assessing privacy risks in mobility data-based ML models. The authors propose a threat model for evaluating privacy risks and provide a comprehensive privacy risk assessment for such models. They also suggest a privacy-preserving solution for point-of-interest recommendation models to mitigate privacy risks. Their contributions include designing four different attacks, such as common location extraction (LOCEXTRACT), training trajectory extraction (TRAJEXTRACT), location-level membership inference attack (LOCMIA), and trajectory-level membership inference attack (TRAJMIA), developing a privacy-preserving training method that protects against data extraction and membership inference attacks aimed at point-of-interest recommendation models. Overall, the authors identify potential privacy risks in mobility data-based machine learning models and propose solutions to address these risks.

**Strengths:**

1) The paper presents a privacy attack suite that is specifically designed for POI recommendation models. This suite consists of data extraction and membership inference attacks. By conducting experiments with real-world mobility datasets, the authors demonstrate the vulnerability of current POI recommendation models to these attacks.

2) The paper investigates the effectiveness of existing defense mechanisms, such as L2 regularization and differential privacy, against the proposed attacks. However, it concludes that these mechanisms have limitations in providing comprehensive protection.

3) The impact of training and attack parameters on attack performance is analyzed. The effect of training epochs on information leakage and the influence of query timestamps on data extraction attack performance are discussed.

**Weaknesses:**

1) The study only evaluates the privacy risks of POI recommendation models in a controlled setting. They didn't provide some insights into how these models might perform in real-world scenarios, where more complex factors are at play.

2) The author did not provide insights on how these risks compare to privacy risks associated with ride-sharing or food-delivery apps.

**Questions:**

1) According to the paper, no definitive defense mechanism can protect against all the proposed attacks simultaneously. Can you please explain why this is the case? Additionally, could you provide insights into the challenges that must be addressed to develop more effective defenses against such attacks?

2) How do the proposed attacks and defense mechanisms compare to existing methods in the literature?

---

> ### Author Response · Authors · 2023-11-16
>
> Dear Reviewer KUtq,
>
> We are grateful for the time you took to review our paper and for your valuable feedback. We would like to present our explanations and responses to your concerns below!
>
> __W1 Only Evalutes the Privacy Risk in A Controlled Setting But Not Real-World Scenarios__
> Our attack is performed on real-world data sets collected by leading social media companies. Since the real-world POI recommendation models are also trained on a similar data source, we believe that our attack on the current dataset should be able to simulate realistic scenarios where the attack might happen.
>
> Also, as mentioned in *Section 2.2*, our attacks do have real-world effects, majorly in two cases where the adversary should have the abilities mentioned in our paper:
>
> (1) In a scenario mentioned by service providers [1,2,3], user data is collected and used to train models, which are then provided as an API to third parties for application development. In this context, a malicious third party might gain access to the confidence scores from the model's output
>
> (2) In a second scenario, a situation arises on the service provider side during the retention period of the training data expires. Still, the model owner retains the model, and an adversary (e.g., a malicious insider of location service providers) can aim to use the model to perform our attack aiming to infer information about the data that has been deleted.
>
> As discussed in the *“Limitation and Future Work”, Section 6*, we also plan to adapt our attack method to some public real-world services with more limitations (i.e. Map APIs which might only provide the predicted label) using methods like label-only attacks.
>
>
> __W2 Comparison to Privacy Risks Associated With Ride-sharing or Food-delivery Apps__
> The prior ride-sharing and food-delivery problems mostly lie in solving the different problems of different privacy stakeholders compared to the POI Recommendation Models.
>
> Firstly, prior ride-sharing and food-delivery research targets the following problems:
>
> (1) The privacy of company-owned data. For example, Lyft and Uber can use the spatial distribution information of their instrumented fleets for better placement and routing [4].
>
> (2) Privacy attacks targeting passengers or companies' service providers, such as drivers. For instance, attackers could steal continuous trajectories and analyze behaviors. However, our paper focuses on attacking a different data format: the user's POIs in location formats, which can be combined into sparse trip trajectories closely related to the user's daily life.
>
> Secondly, the defense mechanisms are different. The studies of Ride-sharing or Food-delivery primarily focus on the data aggregation process, with most relying on secure system design utilizing methods like cryptographic approaches or differential privacy. Our attacks, however, target the privacy leakage of deep learning models, representing a new attack surface. The defense mechanisms for ride-sharing or food delivery are not suitable to defend against our attack.
>
> __Q1 Explanation for Defense Mechanism Not Working and Insights Into More Effective Defenses__
> Thanks for asking this question. This is a great question. We have listed the results of commonly used defense mechanisms in *Appendix E*. These defense mechanisms(e.g. differential privacy), often result in significant drops in utility, primarily due to the nature of the task. The reason is that the utility of existing POI recommendation models heavily depends on memorizing user-specific trajectory patterns from limited user-level data, which lacks sufficient semantic information as in CV/NLP. However, defense mechanisms like differential privacy can diminish the level of memorization.
>
> We believe that to solve this problem, we need a better-designed scheme that can help with generalizing the POI recommendation model. For example, more information in the dataset, such as POI categories or trip purposes, should be included, and this information should be better utilized. Also, we believe that using a pre-trained model on a public dataset and fine-tuning it on a smaller private dataset might help solve this problem. This approach has been successfully used in other tasks in CV [6,7] and NLP [8,9]. This can be a potential future direction, and we plan to work on this in the future.

---

> > ### Author Response · Authors · 2023-11-16
> >
> > __Q2 Comparison of The Attack and Defense Mechanism to Existing Works__
> > As mentioned in *Section 5*, previous mobility attacks are focused on utilizing side-channel attacks to extract sensitive information (e.g., social relationships) during the data aggregation and releasing processes.  We are the first to investigate privacy leakage of mobility data from a DNN-based POI recommendation model.
> >
> > Existing data extraction and membership inference attacks in CV/NLP are insufficient for POI recommendation due to the spatiotemporal nature of the tasks and the lack of consideration of a realistic threat model. In this work, we propose novel attack mechanisms (e.g., spatial-temporal model query) algorithm to promote the privacy attack’s performance on our task.
> >
> > For the defense mechanisms, as mentioned in *Appendix Section F*, since the existing attacks focus on data aggregation and release processes, the prior defense works also concentrate on avoiding these kinds of attacks using methods like k-anonymity, location spoofing, and geo-indistinguishability. In our work, we focus on protecting privacy during the model training steps. We adapt the selective DP method JFT [10] to the POI recommendation task, with clear definitions of sensitive information based on our attack results. Additionally, we utilize other defense mechanisms such as regularization and early stopping mechanisms for comparison. We also test the performance of DP-SGD, which is one of the most widely used mechanisms for protecting deep learning models. We find that it substantially sacrifices the model’s utility in the POI recommendation task, making it unsuitable for this task. For more details, please refer to *Section E in the appendix*.
> >
> > __References:__
> > [1] Chen et al. Curriculum Meta-Learning for Next POI Recommendation.
> > [2] Liu et al. STGIN: Spatial-Temporal Graph Interaction Network for Large-scale POI Recommendation.
> > [3] https://www.localogy.com/2021/03/foursquares-power-play-continues-with-relaunched-places-and-new-api/
> > [4] https://www.ndss-symposium.org/ndss-paper/auto-draft-233/
> > [5] https://ieeexplore.ieee.org/abstract/document/9222176?casa_token=dLG7lPw-wMIAAAAA:07Pku7HQRsdKRLInWukZYSZP7FxMM1nKbVhrNBfkK9KeaANewXabo5fArpfW1bxMjZYT6UkEsQ
> > [6] Yu et al. ViP: A Differentially Private Foundation Model for Computer Vision.
> > [7] Sander et al. Tan without a burn: Scaling laws of dp-sgd.
> > [8] Anil et al. Large-scale differentially private bert.
> > [9] Yu et al.  Differentially private fine-tuning of language models.
> > [10] Shi et al. Just fine-tune twice: Selective differential privacy for large language models.

---

> > > ### Author Response · Authors · 2023-11-17
> > >
> > > Dear Reviewer KUtq,
> > >
> > > Thank you for your valuable feedback on our submission. We have read your comments carefully and have addressed them in our rebuttal. We would be grateful if you could acknowledge if our responses have addressed your concerns. We would also be happy to engage in further discussions if needed. Thank you again for your time and consideration.

---

> > > > ### Author Response · Authors · 2023-11-22
> > > > **Follow Up With Reviewer KUtq**
> > > >
> > > > Dear Reviewer KUtq,
> > > >
> > > > As the discussion period approaches its conclusion, we wish to ensure that all your concerns have been adequately addressed. We would greatly appreciate any additional feedback you may have. Thank you so much for your time and consideration!
> > > >
> > > > Authors of paper 2078

---

### Official Review · Reviewer_sUA6 · 2023-10-27

**Soundness:** 4 excellent
**Presentation:** 4 excellent
**Contribution:** 3 good
**Rating:** 8
**Confidence:** 4

**Summary:**

This paper offers a privacy attack suite (including data extraction and membership inference attacks) on point-of-interest recommendation models, tailored specifically for mobility data. Experiments are performed on three models trained on two distinct datasets. This suite could in principle be used as a privacy auditing tool.

**Strengths:**

The paper is very clearly presented and the thoroughness of experimental results is commendable to the point where I am left with view remaining questions.

The paper demonstrates clear privacy vulnerabilities in POI recommendation systems that should inform future defenses and auditing.

**Weaknesses:**

It would be valuable to understand how attack vulnerability changes with sample size. For very large datasets, where there are generally a larger number of unique users to all locations, does the attack success decline? This theory seems to be somewhat supported by Fig 4.

More detailed descriptions of datasets size and dimensionality would be valuable to understand whether the emulate real-world production systems.

The paper could benefit from explicitly contextualizing how attack performance compares to attack performance for other type of data/models (e.g. text & image are referenced).

**Questions:**

How do we know if the utility privacy trade-off inherent or a limitation of existing DP algorithms? You note the privacy-utility trade-off does not strictly hold in your experiments but can you show that practically acceptable utility and privacy can both be achieved?

Could you explain why having more total check-ins seems to help protect a user against a MIA? This seems surprising in the context of differential privacy where worst-case privacy loss degrades with sensitivity.

Have you explored the feasability of DP synthetic data for this type of application?

---

> ### Author Response · Authors · 2023-11-16
>
> Dear Reviewer sUA6,
>
> We thank you for your valuable feedback and for finding our work meaningful! We would like to respond to your concerns and improve this paper based on your suggestions.
>
> __W1 Effect of Larger Dataset__
> Thank you for pointing this out. We believe the dataset size does not affect our key insights, as our attacks are derived primarily from the task's inherent characteristics. For example, POIs frequently visited by a user are typically more unique to that user (e.g., home address). These locations are always more susceptible to our attack since they are well memorized by the models as shown in Figure 4.
>
> To confirm this, we conducted additional experiments using a larger FourSquare dataset from Tokyo, containing about *4x* more check-ins than the NYC dataset. Our LocExtract and LocMIA attacks on models trained on this dataset are still effective, showing that our attack is agnostic to the dataset scale. Due to the time limitation in the rebuttal period, we have not finished the experiment on the trajectory-level attacks but we will add them to our paper when finished.
>
>
> *LocExtract Result:*
> |          | ASR-1 | ASR-3 | ASR-5 |
> |----------|-------|-------|-------|
> |**4SQ(TKY)** | 40.4% | 66.0% | 72.9% |
> | **4SQ(NYC)** | 33.0% | 57.3% | 64.9% |
>
>
> *LocMIA Result:*
> |          | TPR@%10FPR | AUC  | ACC  |
> |:--------:|:----------:|:----:|:----:|
> | **4SQ(TKY)** |     0.72     | 0.88 | 0.81 |
> | **4SQ(NYC)** |     0.74     | 0.80 | 0.87 |
>
>
> __W2  More Descriptions of Dataset__
> Thanks for pointing out this. We have included some dataset-related information in *Table 3 in the Appendix*, and we will make sure we highlight this information further. Our attack uses real-world datasets that are collected from NYC social media users.
>
> __W3  Attack Comparison with Other Types of Data/Models__
> Thanks for pointing this out! We will add a table for a comparison of MIAs here, as the data extraction attack we proposed is specific to POI recommendation models. As mentioned in [1], LiRA can achieve an *8.4% TPR @0.1%FPR* on image classification models trained on the cipher-10 dataset. In our case, the attack achieved a *20+% TPR @0.1%FPR* for LocMIA on all models trained on the FourSqure dataset, but it’s relatively lower for the Gowalla dataset with a *1+%  TPR @0.1%FPR*. As demonstrated in [1], the difference in the attack success rate can be attributed to the level of memorization in the model.
>
> However, it should be noted that the current POI recommendation model utility still heavily relies on the memorization of the historical visit of a user, which makes the attack relatively easier and leaves the protection of POI-recommendation models more difficult, as we will discuss in the next question.
>
> __Q1  Reason for the Utility Privacy Trade-off and Potential Practical Solution__
> This is a great question/ We have listed the results of commonly used defense mechanisms in *Appendix E*. These defense mechanisms(e.g. differential privacy), often result in significant drops in utility, primarily due to the nature of the task. The reason is that the utility of existing POI recommendation models heavily depends on memorizing user-specific trajectory patterns from limited user-level data, which lacks sufficient semantic information as in CV/NLP. However, defense mechanisms like differential privacy can diminish the level of memorization.
>
> We believe that to solve this problem, we need a better-designed scheme that can help with generalizing the POI recommendation model. For example, more information in the dataset, such as POI categories or trip purposes, should be included, and this information should be better utilized. Also, we believe that using a pre-trained model on a public dataset and fine-tuning it on a smaller private dataset might help solve this problem. This approach has been successfully used in other tasks in CV [6,7] and NLP [8,9]. This can be a potential future direction, and we plan to work on this in the future.

---

> ### Author Response · Authors · 2023-11-16
>
> __Q2  Explanation Why More Total Check-ins Help Protect a User From MIA__
> Thanks for asking! Sensitivity in DP measures the maximum difference between samples in a dataset. When this measurement is at the user level, a user with more check-ins can be more easily distinguished from others, as suggested in [2]. However, this kind of user-level inference attack might not be suitable for this task. The POI recommendation model takes the user ID as input. If such a user does not exist in the dataset, there will be no query results.
> Rather than focusing on a user-level inference attack, our attack aims to infer the membership of individual POIs or trajectories. In this context, if a user has visited many locations, the model may have lower confidence in predicting a single POI, leading to less distinction in confidence scores between members and non-members. This scenario can be likened to the idea that hiding a location among a large number of locations is easier than concealing it among a few. Consequently, the more locations a user has visited, the lower the performance of the attack.
>
> __Q3 Feasibility of DP Synthetic Data for POI Recommendation__
> Synthetic data with DP guarantee is a direction that is worth more investigation. There are some existing studies that discuss the use of synthetic data to protect mobility information, with a primary focus on safeguarding continuous trajectories. For instance, [3] explores the use of differentially private synthetic data to protect POIs. However, they do not address the utility and performance of POI recommendation tasks, as their main target is the private release of data.
>
> Our paper mostly focuses on measuring the privacy leakage from the POI recommendation models. However, Synthetic data generation might require different threat models and attack settings, Thus, we do not include them in our paper. Note that privacy attack for synthetic data is a promising direction that needs independent work to investigate [4, 5]. As a future work, it is interesting to see if the privacy attacks in these works can be applied to mobility data synthetic models like [3].
>
> __References:__
> [1] Carlini et al. Membership Inference Attacks From First Principles.
> [2] Montjoye et al. Unique in the crowd: The privacy bounds of human mobility.
> [3] Rao et al. LSTM-TrajGAN: A Deep Learning Approach to Trajectory Privacy Protection.
> [4] Breugel et al. Membership Inference Attacks against Synthetic Data through Overfitting Detection.
> [5] Hyeong et al. An Empirical Study on the Membership Inference Attack against Tabular Data Synthesis Models.
> [6] Yu et al. ViP: A Differentially Private Foundation Model for Computer Vision.
> [7] Sander et al. Tan without a burn: Scaling laws of dp-sgd.
> [8] Anil et al. Large-scale differentially private bert.
> [9] Yu et al.  Differentially private fine-tuning of language models.

---

> > ### Author Response · Authors · 2023-11-17
> >
> > Dear Reviewer sUA6,
> >
> > Thank you for your acknowledgment and the valuable feedback on our submission. Your comments have been very helpful, and we have incorporated corresponding explanations in the manuscript as well as provided some here. We would appreciate it if you could confirm whether our responses adequately address your comments. Additionally, we are open to and welcome any further discussions if needed. Thank you once again for your time and consideration.

---

> > > ### Author Response · Authors · 2023-11-22
> > > **Follow Up With Reviewer sUA6**
> > >
> > > Dear Reviewer sUA6,
> > >
> > > As the discussion period approaches its conclusion, we wish to ensure that all your concerns have been adequately addressed. We would greatly appreciate any additional feedback you may have. Thank you so much for your time and consideration!
> > >
> > > Authors of paper 2078

---

### Official Review · Reviewer_1v9T · 2023-11-07

**Soundness:** 2 fair
**Presentation:** 3 good
**Contribution:** 2 fair
**Rating:** 5
**Confidence:** 4

**Summary:**

This paper evaluates the privacy risks of POI recommendation models by introducing an attack suite and conducts extensive experiments to demonstrate the effectiveness of these attacks. Additionally, it analyzes which types of mobility data are vulnerable to the proposed attacks and further adapts two mainstream defense mechanisms to the task of POI recommendation.

**Strengths:**

S1. The scenario of this paper, POI recommendation, has real-world applications.

S2. The paper proposed several attack methods.

S3. Extensive experiments are conducted on two real datasets.

**Weaknesses:**

W1. The motivation of this work is not convincing enough.

W2. The definition of sensitive information is unclear.

W3. Experiments are inadequate and some insights are not surprising.

W4. There are some typos in this paper. For example, at line 16 in algorithm 4,   $f_out$ should be $f_\theta$.

**Questions:**

Q1. The definition of sensitive information and privacy guarantee should be formally defined and well justified. Then, it may become meaningful to conduct adversary attacks.
Q2. There have been quite a few studies on protecting spatial/location/trajectory privacy. However, most of them was not reviewed/evaluated by this paper. Thus, it was uncertain whether existing privacy preservation mechanisms could help on the mentioned limitation of POI recommendation.
Q3. Take private spatial data publish as an example. Based on GDPR, a LBS platform can only collect user’s check-in data that has been well protected (e.g., by differential privacy). Under this practical setting, deriving the platform’s data will not leak the sensitive information of users, which makes the attacker model proposed in this work less meaningful.
Q4. I am also curious: if the input data has been well protected by existing privacy mechanism and then trained by POI recommendation model, is there any sensitive information leakage?
Q5. Since privacy preserving learning has been well studied in recent years, it is sometimes possible to extend existing POI recommendation models with privacy guarantee (e.g., by adding differential privacy noise in the gradients). Does this fact significantly change the main insight?
Q6. In Appendix E, the epsilon setting of DP-SGD is a little large. Please provide more justifications.
Q7. Both datasets are a little outdated and relatively smaller-scale than the current LBS platform. However, the major insights are strongly related to the data sparsity. Maybe, it would be better to conduct experiments on large-scale datasets.
Q8. Why does the curve in Figure 5(b) first rise and then drop when the number of POI increases?
Q9. It is mentioned that k is usually 1, 5, and 10 when using top-k to measure accuracy in page 3. However, only 1, 3, and 5 were tested in the experiment as shown in Figure 1. What is the rationale of this experimental setting?

---

> ### Author Response · Authors · 2023-11-16
>
> Dear Reviewer 1v9T,
>
> We thank you for taking the time to review our paper and provide valuable feedback and questions. We would like to present our explanations below and hope they can address your concerns.
>
> __W3  Experiment and Insights Are Not Surprising__
> We are willing to adjust the problems or explain our experiments in more detail. Can you please provide more suggestions on the experiments that could further improve the experiment?
>
> __W4  Typos In The Paper__
> Thank you for pointing out the typos. We will fix them in the next version.
>
>  __Q1\W2  Need Formally Define Sensitive Information and Privacy Guarantee__
> We have included the notations and definitions of the objectives for each attack in *Appendix Table 2*, accompanied by their respective mathematical definitions. For your convenience, the corresponding definition table is also attached below for reference.
>
> | **Attack**     | **Adversary Objective** | **Adversary Knowledge** |
> |----------------|-------------------------|-------------------------|
> | `LocExtract`   | Extract the most frequently visited location $l$ of a target user $u$ | -- |
> | `TrajExtract`  | Extract the location sequence of a target user $u$ with length $n$: $x_L=$\{$l_0, \dots, l_{n-1}$\} | Starting location $l_0$ |
> | `LocMIA`       | Infer the membership of a user-location pair ($u$,$l$) | Shadow dataset $D_{\mathrm{s}}$ |
> | `TrajMIA`      | Infer the membership of a trajectory sequence $x_T=$\{$(l_0,t_0),\dots,(l_n,t_{n})$\} | Shadow dataset $D_{\mathrm{s}}$ |
>
>
> Our definitions of sensitive information in our attack suite are similar to previous works studying privacy risks on general machine learning models [1,2]. We appreciate it if you could provide additional suggestions and clarifications on improving the definition of sensitive information.
>
>
>
>
> __Q2  Need Defense-Related Literature Survey__
>  We have listed the attack-related works in *section 5* and defense-related works in *appendix F*.
>
> As mentioned in *Section 5*, prior research primarily focused on exploiting side channels such as social relationships and historical trajectories to extract sensitive information during the data aggregation and release processes. However, these studies have not yet concentrated on potential leakages through deep neural networks. Given that more recent LBS relies on deep neural networks, which present a new attack surface for adversaries seeking to access private user information, these earlier attack studies are insufficient. We are the first to investigate privacy leakages in POI recommendation models based on deep neural networks. Our work highlights the need for a privacy-preserving ML-based POI recommendation system in both industry and academia.
>
> As for defense, prior works primarily focused on data aggregation and release processes tailored to the attack mentioned above. Since our paper focuses on evaluating the privacy leakage associated with the model, we only focus on the defense mechanisms during the training process, which is widely ignored in prior works. We also will discuss more defenses, e.g., LDP) in the related work section.
>
> We also appreciate it if you could identify some missing related work, and we will cite and compare them accordingly.
>
> __Q3/W1  GDPR and Attack Motivation__
> We would like to emphasize that the threat model of the paper is well-motivated, and the privacy study of the work is urgently needed, as required by GDPR.
>
> First of all, GDPR does not mention requirements for using specific protection mechanism mechanisms like differential privacy in collecting user data like location data. GDPR only requires “the collected content is needed, minimized, and stored safely” in articles *#5 and #25* [3,4]. Since the potential threats in POI recommendation models have not been well studied, companies currently have not adopted adequate protection mechanisms when receiving the data [8,9]. Even if there are protection mechanisms like differential privacy on training data, existing works lacks measurement study to quantify whether those data protection mechanisms are properly applied to the data. Our work could bridge this gap by using our proposed attack suite to perform such kind of measurement.
>
> Moreover, no prior work has studied how POI recommendation models leak sensitive information about users’ data, and our work is the first one to study this problem. This is an important problem since POI recommendation models could unintentionally leak user’s sensitive information.

---

> > ### Author Response · Authors · 2023-11-16
> >
> > __Q4  Effect of Using Differential Privacy On Training Data__
> > This is a good question. Adding noises with differential privacy guarantees to the training data before model training can help mitigate privacy leakage. However, it comes with a significant utility drop for POI recommendation models. In our study, we also apply a variant of differential privacy called geo-indistinguishability [5] in *section E*. By only protecting part of the training data, we already observed a significant utility drop, as shown in Figure 18.
> >
> > Besides, existing works [6,7] that utilize Local Differential Privacy (LDP) to protect training data are insufficient in protecting POI recommendation models. In particular, [6] has to aggregate all the POIs in large regions to form some new (region-level) POIs to achieve reasonable performance, which alters the definition of the original task. While [7] provides POI recommendations for original POIs using an LDP-protected dataset, they show that adding LDP to the training data results in poor utility with a precision of only 0.08. To this end, we highlight that purely using differential privacy on training data might not be suitable for direct application in our task and require better design.
> >
> > __Q5  Effect of Using DP-SGD During Model Training__
> >  We conduct the corresponding experiments by adding differential privacy noise in the gradients (e.g., DP-SGD, Selective-DP)  and demonstrate their effects in section *Appendix E*. From the result, we can see that although using these defense mechanisms can mitigate privacy loss, they also cause significant utility loss. That’s why we highlighted that the current defense mechanism might not be enough, and we have listed stronger defense mechanisms tailored to this problem in the section of future work. One promising direction to improve the defense is to use the pre-trained model on a public dataset and fine-turned on a smaller private dataset, which might help with solving this problem.
> >
> >
> > __Q6  Epsilon Selection of DP-SGD__
> > For DP-SGD, the top-10 accuracy is dropped to 4.97% when the mechanism satisfies (5, 0.001)-DP, which already makes the task not practical. We do not show the result of an epsilon of 0.5 in the paper. This is because when epsilon is set to 0.5, there is a significant drop in utility; specifically, the top-10 accuracy decreases from 71% (when not protected) to below 1%. which makes the protection less practical. We also show the result of epsilon of 1 and 5 for the selective DP in *Appendix E*. and do not further show smaller epsilons which makes the protection less practical.
> >
> > The significant utility drop happens because the POI recommendation requires the model to memorize user-specific patterns from very limited user-level training data. Thus, the training is quite sensitive to the noises introduced by DP-SGD. Adding semantics information to the task or finetuning on pre-trained models might mitigate this problem.
> >
> > __Q7  Need Larger-scale Dataset__
> > Thanks for the suggestion!  We currently use standard datasets and SOTA models for POI recommendations in academia. The FourSquare NYC dataset we used is also used in the latest works used in the literature [10,11]. We believe the dataset size does not affect our key insights, as our attacks are derived primarily from the task's inherent characteristics. For example, POIs frequently visited by a user are typically more unique to that user (e.g., home address). These locations are always more susceptible to our attack since they are well memorized by the models as shown in Figure 4.
> >
> > To confirm this, we conducted additional experiments using a larger FourSquare dataset from Tokyo, containing about *4x* more check-ins than the NYC dataset. Our LocExtract and LocMIA attacks on models trained on this dataset are still effective, showing that our attack is agnostic to the dataset scale. Due to the time limitation in the rebuttal period, we have not finished the experiment on the trajectory-level attacks but we will add them to our paper when finished.
> >
> > *LocExtract Result:*
> > |          | ASR-1 | ASR-3 | ASR-5 |
> > |----------|-------|-------|-------|
> > | **4SQ(TKY)** | 40.4% | 66.0% | 72.9% |
> > | **4SQ(NYC)**| 33.0% | 57.3% | 64.9% |
> >
> > *LocMIA Result:*
> > |          | TPR@%10FPR | AUC  | ACC  |
> > |:--------:|:----------:|:----:|:----:|
> > | **4SQ(TKY)** |     0.72     | 0.88 | 0.81 |
> > | **4SQ(NYC)** |     0.74     | 0.80 | 0.87 |

---

> ### Author Response · Authors · 2023-11-16
>
> __Q8  Explanation of Fluctuation in Figure 5(b)__
> We believe that the fluctuation in Figure 5(b) can be attributed to two main reasons.
>
> (1) We hypothesize that when the trajectory is very short, it may be more easily influenced by other factors, such as increased location sharing. Additionally, as mentioned in our paper, many other characteristics can impact the performance of the attack. So they can also contribute to the fluctuations.
>
> (2) When the trajectory is longer, it is more difficult to be memorized by the model. Since our attack considers all location points in the trajectory, it is more likely some locations are less well memorized than others and cause a drop in attack success rate. This also explains why the overall trend is that longer trajectories are less vulnerable to our MIA, and a similar result can also be observed in Figure 17, where we conduct a study from the attack parameter selections.
>
> __Q9  Different Selection of “topk-accuracy” and ”topk-asr”__
> Thanks for pointing out this notation confusion. The *topk-accuracy* of (1,5,10) on page 3 is in evaluating the performance of the POI recommendation task itself, whereas in the later part of the paper, we also use the same setting (and mostly k=10) to evaluate the utility of the model with/without protection.
>
> However, the result in Figure 1 is actually *topk-ASR* for the data extraction attacks, where a smaller k makes more sense as it represents the reliability of the attack. That’s why we use  k=1/3/5 to evaluate the attack results. We will state this clearly in the next version of the paper.
>
> __References:__
> [1] Liu et al. ML-Doctor: Holistic Risk Assessment of Inference Attacks Against Machine Learning Models.
> [2] Carlini et al. Membership Inference Attacks From First Principles.
> [3] https://gdpr.eu/article-5-how-to-process-personal-data/
> [4] https://gdpr-info.eu/art-25-gdpr/
> [5] Andrés et al. Geo-indistinguishability: Differential privacy for location-based systems.
> [6] Bao et al. Successive Point-of-Interest Recommendation With Personalized Local Differential Privacy.
> [7] Xu et al. An efficient privacy-preserving point-of-interest recommendation model based on local differential privacy.
> [8] Chen et al. Curriculum Meta-Learning for Next POI Recommendation.
> [9] Liu et al. STGIN: Spatial-Temporal Graph Interaction Network for Large-scale POI Recommendation.
> [10] Luo et al. Timestamps as Prompts for Geography-Aware Location Recommendation.
> [11] Yang et al. GETNext: Trajectory Flow Map Enhanced Transformer for Next POI Recommendation.

---

> ### Author Response · Authors · 2023-11-17
>
> Dear Reviewer 1v9T,
>
> Thank you very much for taking the time to review our submission. We deeply appreciate your thorough feedback and evaluation of our work. In our rebuttal response, we have carefully considered your comments. We would be grateful for your acknowledgment of our responses and your feedback on whether they address your concerns. We are happy to engage in further discussions if needed. Once again, thank you for your invaluable input and attention to our work.

---

> > ### Author Response · Authors · 2023-11-22
> > **Follow up with Reviewer 1v9T**
> >
> > Dear Reviewer 1v9T,
> >
> > As the discussion period approaches its conclusion, we wish to ensure that all your concerns have been adequately addressed. We would greatly appreciate any additional feedback you may have. Thank you so much for your time and consideration!
> >
> > Authors of paper 2078

---

> > > ### Comment · Reviewer_1v9T · 2023-11-23
> > > **Response to the author feedback**
> > >
> > > Thank you for the detailed respnse and sorry for the late reply. I have raised my score.

---

> > > > ### Author Response · Authors · 2023-11-23
> > > > **Thank You!**
> > > >
> > > > Dear Reviewer 1v9T,
> > > >
> > > > We are deeply grateful for your constructive feedback and recognition. Your comments are helpful and if you have any additional comments, please feel free to share them with us. Once again, thank you for dedicating your time and expertise to review our paper. Your contribution is greatly appreciated.
> > > >
> > > > Sincerely,
> > > >
> > > > The Authors of Paper 2078

---

### Official Review · Reviewer_iRwY · 2023-11-07

**Soundness:** 3 good
**Presentation:** 3 good
**Contribution:** 2 fair
**Rating:** 6
**Confidence:** 2

**Summary:**

This paper proposes data extraction and membership inference attacks to POI recommendations involving location data. The experiments are conducted in two datasets and the empirical results show the effectiveness of the proposed attacks.

**Strengths:**

1. The research problem is important.
2. The paper analyzes the factors in the data that affect the attack performance.
3. The proposed attacks for data extraction and membership inference attacks are simple yet effective.

**Weaknesses:**

1. The appropriate baselines are missing. Can the existing data extraction or membership inference attacks be applied to the POI recommendation models, e.g., [1]?
2. There is no qualitative result analysis on the data extraction attacks. It would be better if the authors could conduct these analyses on the data extraction attacks.
3. The threat models assume that the adversaries are capable of accessing the confidence scores, which makes them impractical. In practice, the model owner only releases the final result to the users. In this case, whether the proposed attacks are still effective is unknown.




[1] R. Shokri, M. Stronati, C. Song and V. Shmatikov, "Membership Inference Attacks Against Machine Learning Models," 2017 IEEE Symposium on Security and Privacy (SP), San Jose, CA, USA, 2017, pp. 3-18, doi: 10.1109/SP.2017.41.

**Questions:**

N/A

---

> ### Author Response · Authors · 2023-11-16
>
> Dear Reviewer iRwY,
>
> We thank you for taking the time to review our paper and for acknowledging the importance of our research problem. We attach our responses to the concerns below and hope to adequately address them.
>
> __W1  Baselines Missing__
> Thanks for the suggestions! We believe that the existing inference attacks can work, but we have not included them in the paper for the following reasons:
>
> (1) Since our paper primarily focuses on evaluating the privacy risks in POI recommendation models. Thus, we select the SOTA attack – LiRA [2], as it has been shown to outperform other MIAs, including [1].
>
> (2) Our main contribution is in adapting the existing methods to the POI context. Our proposed mechanism should also work across different attack approaches like [1] as well, as they both utilize similar techniques like shadow model training.
>
> __W2  Qualitative Result__
> We will add some qualitative result analysis on the data extraction attack and corresponding visualization (e.g., map visualization) to help readers better understand the analyses of the attack. We also appreciate it if you could provide suggestions on the qualitative study that we can conduct.
>
> __W3  Threat Model and Confidence Scores__
>  Thanks for asking. We define the corresponding threat model and realistic cases in *section 2.2 (Adversary Knowledge)*. We highlight the scenarios below:
>
> (1) Instead of focusing on publicly available models like Google Maps, which might only provide the final predicted label, we are targeting the scenarios of service providers [3,4,5], who have mentioned in their service that collected user data and trained models, then providing these trained models as services to 3rd party to build applications. Under this setting, a malicious 3rd party might have the confidence scores from the model output.
>
> (2) Also, we have mentioned another scenario where the retention period of the training data expires. Still, the model owner keeps the model, and an adversary (e.g., a malicious insider of location service providers) can aim to use the model to perform our attack to infer about the deleted data.
>
> __References:__
> [1] Shokri et al. Membership Inference Attacks against Machine Learning Models
> [2] Carlini et al. Membership Inference Attacks From First Principles
> [3] Chen et al. Curriculum Meta-Learning for Next POI Recommendation
> [4] Liu et al. STGIN: Spatial-Temporal Graph Interaction Network for Large-scale POI Recommendation
> [5] https://www.localogy.com/2021/03/foursquares-power-play-continues-with-relaunched-places-and-new-api/

---

> > ### Author Response · Authors · 2023-11-17
> >
> > Dear Reviewer iRwY,
> >
> > We are grateful for the insightful feedback you have provided on our submission. We have carefully considered and addressed your comments in our rebuttal. We would appreciate it if you could inform us whether our responses adequately address your concerns. We are also open to further discussion or providing additional explanations as needed. Thank you once again for your time and consideration.

---

> > > ### Author Response · Authors · 2023-11-22
> > > **Follow Up With Reviewer iRwY**
> > >
> > > Dear Reviewer iRwY,
> > >
> > > As the discussion period approaches its conclusion, we wish to ensure that all your concerns have been adequately addressed. We would greatly appreciate any additional feedback you may have. Thank you so much for your time and consideration!
> > >
> > > Authors of paper 2078

---

### Author Response · Authors · 2023-11-16
**General Response**

We thank all the reviewers for their helpful feedback and suggestions. We have listed the common questions here and also responded to the reviewers' comments individually.

__Q1: Realistic Threat Model and Attacker Ability *(Reviewer iRwY W3 & Reviewer KUtq W1)*__

We define the threat model for each attack and realistic cases in *section 2.2 (Adversary Knowledge)*. We highlight the scenarios below:

(1) We are targeting the scenarios of service providers [3,4,5], who have mentioned in their service that collected user data and trained models, then provided these trained models as services to 3rd party to build applications. Under this setting, a malicious 3rd party might have the confidence scores from the model output.

(2) Also, we have another scenario where the retention period of the training data expires. Still, the model owner keeps the model, and an adversary (e.g., a malicious insider of location service providers) can aim to use the model to perform our attack to infer about the deleted data.

These scenarios are recognized in literature [1,2] and industry [3,4]. Under these two real-world scenarios, the adversary will have the ability mentioned in our threat model including the access to the confidence score.

__Q2: Size of The Dataset *(Reviewer 1v9T Q7 & Reviewer sUA6 W1)*__

Following the literature [9,10], we evaluate state-of-the-art POI recommendation models built on the standard datasets, including FourSquare NYC and Gowalla. We believe the dataset size does not affect our key insights, as our attacks are derived primarily from the task's inherent characteristics. For example, POIs frequently visited by a user are typically more unique to that user (e.g., home address). These locations are always more susceptible to our attack since they are well memorized by the models as shown in Figure 4.

To confirm this, we conducted additional experiments using a larger FourSquare dataset from Tokyo, containing about 4x more check-ins than the NYC dataset.  Our LocExtract and LocMIA attacks on models trained on this dataset are still effective, showing that our attack is agnostic to the dataset scale.

__Q3: Why the Current Defense Mechanisms Are Not Working and Potential Future Directions *(Reviewer sUA6 Q1 & Reviewer KUtq Q1 & Reviewer 1v9T Q4/Q5)*__

We have listed the results of commonly used defense mechanisms in *Appendix E*. These defense mechanisms (e.g., differential privacy), often result in significant drops in utility, primarily due to the nature of the task. The reason is that the utility of existing POI recommendation models heavily depends on memorizing user-specific trajectory patterns from limited user-level data, which lacks sufficient semantic information as in CV/NLP. However, defense mechanisms like differential privacy can diminish the level of memorization.

We believe that to solve this problem, we need a better-designed scheme that can help with generalizing the POI recommendation model. For example, more information in the dataset, such as POI categories or trip purposes, should be included, and this information should be better utilized. Also, we believe that using a pre-trained model on a public dataset and fine-tuning it on a smaller private dataset might help solve this problem. This approach has been successfully used in other tasks in CV [5,6] and NLP [7,8]. This can be a potential future direction, and we plan to work on this in the future.



__References:__
[1] Chen et al. Curriculum Meta-Learning for Next POI Recommendation
[2] Liu et al. STGIN: Spatial-Temporal Graph Interaction Network for Large-scale POI Recommendation
[3] https://www.localogy.com/2021/03/foursquares-power-play-continues-with-relaunched-places-and-new-api/
[4] https://venturebeat.com/data-infrastructure/how-foursquare-helps-enterprises-drive-positive-results-with-geospatial-technology/
[5] Yu et al. ViP: A Differentially Private Foundation Model for Computer Vision
[6] Sander et al. Tan without a burn: Scaling laws of dp-sgd.
[7] Anil et al. Large-scale differentially private bert.
[8] Yu et al.  Differentially private fine-tuning of language models.
[9] Yang et al. GETNext: Trajectory Flow Map Enhanced Transformer for Next POI Recommendation
[10] Luo et al. Timestamps as Prompts for Geography-Aware Location Recommendation

---

### Meta-Review · Area_Chair_chLL · 2023-12-05

**Metareview:**

While there were a few positive scores for the paper, the overall sentiment for the paper was still on the negative side. Two main prevalant concerns that persisted: a) lack of motivation for the problem, and b) the conclusions from the paper are unsurprising (especially from Reviewer 1v9T). This resulted in the overall decision for the paper.

**Justification For Why Not Higher Score:**

The concerns from the reviewers persisted even after the rebuttal.

**Justification For Why Not Lower Score:**

NA

---

### Decision · Program_Chairs · 2024-01-16

Reject